# Multiplex, high-throughput method to study cancer and immune cell mechanotransduction
Abigail R. Fabiano ⓘ , Spencer C. Robbins, Samantha V. Knoblauch, Schyler J. Rowland, Jenna A. Dombroski & Michael R. King ⓘ ✉

Studying cellular mechanoresponses during cancer metastasis is limited by sample variation or complex protocols that current techniques require. Metastasis is governed by mechanotransduction, whereby cells translate external stimuli, such as circulatory fluid shear stress (FSS), into biochemical cues. We present high-throughput, semi-automated methods to expose cells to FSS using the VIAFLO96 multichannel pipetting device custom-fitted with 22 G needles, increasing the maximum FSS 94-fold from the unmodified tips. Specifically, we develop protocols to semi-automatically stain live samples and to fix, permeabilize, and intracellularly process cells for flow cytometry analysis. Our first model system confirmed that the pro-apoptotic effects of TRAIL therapeutics in prostate cancer cells can be enhanced via FSS-induced Piezo1 activation. Our second system implements this multiplex methodology to show that FSS exposure (290 dyn cm$^{-2}$) increases activation of murine bone marrow-derived dendritic cells. These methodologies greatly improve the mechanobiology workflow, offering a high-throughput, multiplex approach.

Cancer remains the leading cause of death worldwide, and the mortality rate is higher in men compared to women[1]. The five-year survival rate for men with distant prostate cancer (PCa) is 32%[2]. Reports frequently state that more than 90% of cancer-related deaths involve patients who present metastatic cancer, or cancer that has spread to other parts of the body[3]. However, the precise statistic is unknown[4]. Cancer metastasis is a multi-step process in which cancer cells experience (1) invasion into the surrounding tumor stroma, (2) intravasation into the bloodstream, (3) convective transport through the circulation, (4) extravasation out of the bloodstream, and (5) colonization at a secondary site[5]. Cancer cells that circulate in the bloodstream during this cascade of events are referred to as circulating tumor cells (CTCs). CTCs endure harsh physiological conditions such as fluid shear stresses (FSS) that are not present in the primary tumor environment, while traveling through the circulatory system[6–10]. Elucidating how CTCs alter their mechanobiology to adapt to and survive these conditions remains largely unknown. This understanding is essential, given that less than 0.01% of CTCs eventually may grow into macroscopic metastases[5,6].

Mechanotransduction occurs when cells sense and translate external forces or signals, such as FSS, into a biochemical response, and has gained much attention in the field of cancer research[11]. Physiological FSS in humans ranges from 0.5 to 30.0 dyn cm$^{-2}$ (0.05–3 Pa) in veins and arteries, in which shear rates typically range from 160 s$^{-1}$ to 900 s$^{-1}$ [12,13]. FSS can reach 100 dyn cm$^{-2}$ (10 Pa) at valve leaflets within the heart[7,14,15] and can exceed these intensities in cancer patients who present stenosis[16–18]. FSS can also reach peak values over 1000 dyn cm$^{-2}$ (100 Pa) at vessel bifurcations or turbulent flows near the heart[19–24]. CTCs may also experience even higher FSS in certain pathological conditions or in patients with mechanically implanted heart valves[22,25,26]. Current methods to expose cells to FSS consists of single-chamber cone-and-plate devices such as viscometers to produce low-intensity FSS, and syringe pump apparatuses coupled with 30 G needles to recreate high-intensity FSS[20,27,28]. Additionally, peristaltic pumps coupled with microfluidic systems have been used to model FSS ranging up to ~60 dyn cm$^{-2}$ to investigate the effects of FSS on cellular characteristics such as viability and proliferation[29–31].

These methods (i.e. viscometers) have been employed to study how physiologically relevant mechanical stimuli enhance the effectiveness of anti-cancer therapies and can be used to amplify ex vivo immune cell inflammatory responses[20,28,32]. For example, our previous work used cone-and-plate viscometers to expose breast, prostate, and colon cancer cells to FSS[28]. These studies delineated how FSS as a form of mechanical stimulus sensitized these cancer cells to tumor necrosis factor-α (TNF-α)-related apoptosis-inducing ligand (TRAIL)-mediated apoptosis. Soluble TRAIL has

Department of Biomedical Engineering, Vanderbilt University, 2414 Highland Ave, Nashville, TN 37212, USA.
✉e-mail: mike.king@rice.edu

been tested in the clinic, but has shown reduced effectiveness due to its short half-life in circulation and risk of liver toxicity[33]. These enhanced pro-apoptotic effects observed by our group occur through mechanical activation of Piezo1, a mechanosensitive ion channel (MSC)[28]. TRAIL binds to death receptors 4 and 5 on cancer cells, and concurrent activation of Piezo1 produces a synergistic mechanism to activate intrinsic apoptosis through calcium-induced mitochondrial dysfunction[28].

Limitations prevail with these established methods for applying physiological FSS to cells, by offering a restricted parameter space and are generally implemented as single channel fluid systems in which the array of conditions tested simultaneously is limited. When using cone-and-plate devices, it is also difficult to obtain real-time biochemical data given that this is a closed system[34,35]. The syringe pump apparatus requiring reloading of the syringe after each shear pulse, lacks automation and is labor-intensive[3,20]. Methods involving a syringe pump coupled with a microfluidic system have been developed but have been tested primarily at lower FSS intensities (i.e. 5 dyn cm$^{-2}$) and lower fluid velocities (0.1 mL min$^{-1}$) and may contain regions of dead volume[36–38]. However, microfluidic systems are lower throughput by nature, are more costly and time-consuming to maintain production, and may generate unsteady flow conditions in which air bubbles or obstructions in tubing can impact experimental results[29–31,37,39]. Implementing orbital shakers is not a preferred method since the FSS that the cells are exposed to varies greatly within the system, and are very low in magnitude within macroscopic containers[34]. The downstream analysis following FSS exposure also requires the researcher to transfer samples from the experimental devices to centrifuge tubes or well plates.

To address these limitations, we demonstrate how the VIA-FLO96, a commercial, multichannel, semi-automated pipetting device can be tailored to provide an alternate means of studying physiological FSS by permitting higher-throughput protocols. With this device, researchers can further delineate how one can exploit the intrinsic force supplied by the circulatory system when targeting cancer cells and activating immune cells for therapeutic applications. To raise the maximum FSS of the VIAFLO96 33-fold from 8.79 dyn cm$^{-2}$ to 290 dyn cm$^{-2}$, small bore, luer-fit needles were joined with the pipetting head. Briefly, programs were developed using the VIALINK software to shear cells by generating repeated mixing cycles, ranging from 500–10,000 mixes (20 ~ 400 min runtime) for the PCa studies, referred to as "shearing cycles". PCa cells were exposed to FSS in a 96-well format, while exposed to TRAIL. Cell death and mitochondrial depolarization were assessed using manual and semi-automated methods by taking advantage of the multiplex capabilities of the device to prepare samples for flow cytometry to determine the extent of cellular responses.

In addition to CTCs, immune cells, such as dendritic cells (DCs) and T cells are exposed to biophysical forces in the circulatory system, and less is known regarding how mechanical forces affect their function and communication. DCs are innate immune cells and function as potent antigen-presenting cells, to prime T cells to promote anti-tumor function[40–42]. Exposing DCs to FSS in vitro has been shown to promote growth, maturation, and cell cycle progression, while increasing co-stimulatory molecule expression, including MHC-II, CD80, CD40, and CD86[42,43]. To expand the capabilities of the semi-automated methodologies presented, we also examined the cellular response of murine bone marrow-derived dendritic cells (BMDCs) to FSS exposure using the VIAFLO96 system, to advance previous studies which exposed BMDCs to low-intensity FSS (0.2–5.0 dyn cm$^{-2}$) ex vivo[42,43]. We developed a multiplex VIALINK program to permeabilize, fix and stain cells with multiple antibodies simultaneously for flow cytometry assessment.

The VIAFLO96 is necessary to utilize the custom programs included with this article, however, the ability to employ such instruments with more customized or individualized experiments is nearly limitless. The purpose of these methods is to demonstrate how the capabilities of commercially available, multiplex technology can be advanced to provide an alternative method to study cancer and immune cell mechanotransduction. For the cancer cell studies, we utilized weakly metastatic LNCaP cells and highly metastatic PC3 cells as a model[3,44,45]. With this goal, we demonstrate how mechanical stimuli administered over a wide range of duration for multiple cell lines at once can be combined with the cancer-specific drug TRAIL to enhance therapeutic efficacy at an elevated FSS intensity. We also demonstrate the versatility of these methods by demonstrating how the response of primary BMDCs is enhanced via FSS without the addition of an exogenous DC activation agent. These studies encompass the potential for these methods to be translated to scale up clinical practices when treating cancer by taking advantage of the multiplex capabilities this device enables, to process an array of sample types simultaneously.

## Results
### Computational analysis estimates average fluid shear stress generated by unmodified pipette tips
Here we present a semi-automated protocol to study cancer cell mechanotransduction in response to physiological FSS, by adapting the VIAFLO96 to overcome limitations with preestablished FSS methods. The time-averaged local shear stress experienced by a typical cell in the narrowing region of the INTEGRA pipette tips was estimated by numerically integrating in MATLAB the equations governing the flow of a Newtonian fluid (Fig. 1). The FSS (Fig. 1a, c) and velocity (Fig. 1b, d) were modeled as a function of distance traveled through the narrowing region of the pipette tips based on the respective length.

This maximum FSS on average was resolved to be 3.07 and 8.79 dyn cm$^{-2}$ for the "standard" and "long" INTEGRA pipette tips, respectively, corresponding to the fastest VIAFLO96 operating flow rate of 295 µL s$^{-1}$. At this flow rate, the transit time through this region was determined to be 571 ms for the "standard" and 279 ms for the "long" pipette tips. With the pipette tips narrowing from X = 0 to L, this means that a typical cell is only experiencing a maximum FSS of 432 dyn cm$^{-2}$ where X = L of the "long" pipette tip (at 295 µL s$^{-1}$) for a brief period before exiting. The mean transit time and average FSS for the "standard", "long" and "wide bore" INTEGRA pipette tips at all ten flow rates at which the VIAFLO96 can operate are listed in Table S1, and the FSS as a function of the distance traveled in the narrowing region of the pipette tips at the minimum (11.8 µL s$^{-1}$) and maximum flow rates (295 µL s$^{-1}$) is compared in Fig. S1. The maximum FSS magnitude was further increased by joining 22 G needles to latch onto the "long" INTEGRA pipette tips to operate with the VIAFLO96 (Fig. 2).

Briefly, the "long" pipette tips were trimmed at the start of the narrowing region of the tip, and the 22 G needle was secured to the "long" tips using epoxy and Rhino Glue (Fig. 2a–g). Each time the 22 G needles were autoclaved after use, their weight was recorded to ensure that no major deterioration effects occurred due to this process (Fig. 2h). Generally, these custom-made 22 G needle-tips were autoclaved and re-used 5–7 times for experiments, without any evidence of leaking observed. The average FSS a cell experiences during a single transit through the needle was calculated based on the Hagen-Poiseuille model of fluid flow in a pipe of constant diameter, and was determined to be 290 dyn cm$^{-2}$ (29.0 Pa) (Table S2)[3,20] with a mean transit time of 5.69 ms. The FSS in a conduit varies linearly with radial position, in which the maximum FSS is experienced at the bounding surface of the needle, consistent with the FSS that CTCs experience in the veins and arteries during circulation when traveling in the near-wall region[3,20]. The minimum FSS that each cell experiences while traversing through the needle was estimated to be 19.8 dyn cm$^{-2}$, assuming a PCa cell radius of 9.31 µm[20,46]. The transit time through elevated pulses within the heart valves is estimated to be 1–30 ms, thus our modified tips successfully recapitulate these physiological conditions[47]. The average FSS and transit time for a cell in the upper region of the "long" pipette tip (Fig. S2b(1)) was estimated to be 0.740 dyn cm$^{-2}$ (0.0740 Pa) for comparison. In the

**Fig. 1 | Numerical calculation of a cell traversing through a narrowing pipette tip. a** The average fluid shear stress and (**b**) average velocity that a cell experiences while traveling the length of the "standard" INTEGRA pipette tips. **c** The average fluid shear stress and (**d**) average velocity a cell experiences traveling the length of the "long" INTEGRA pipette tips. X/L represents the dimensionless length traveled through the narrowing region of the pipette tip. The dashed black line (**a**, **c**) represents the average fluid shear stress at the maximum speed of 295 μL s⁻¹. Each solid, colored line corresponds to the VIAFLO96 flow rates from 11.8 to 295 μL s⁻¹.

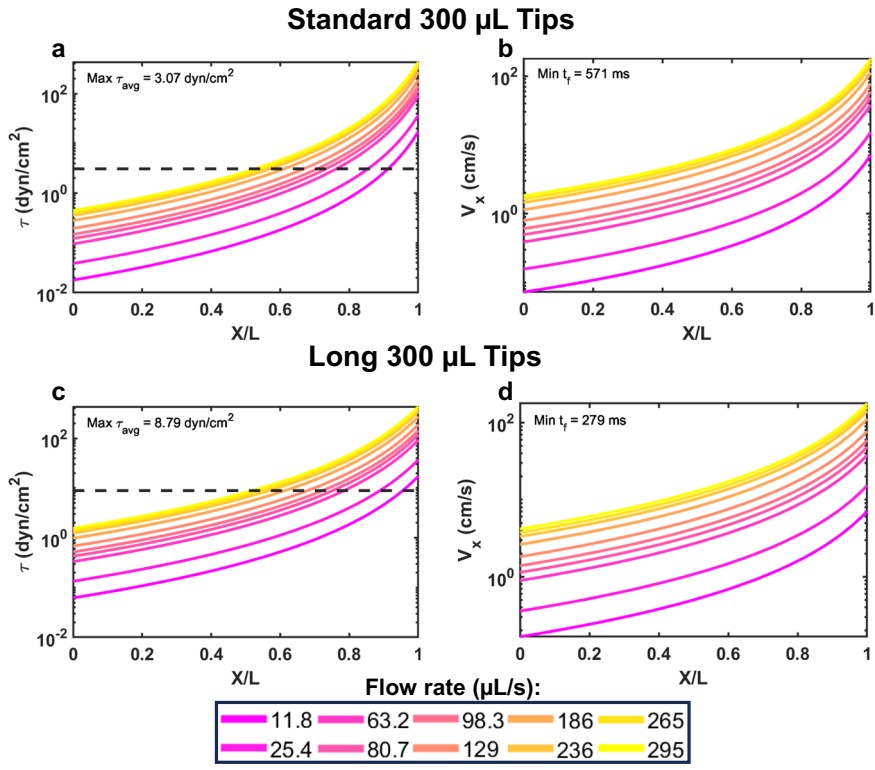

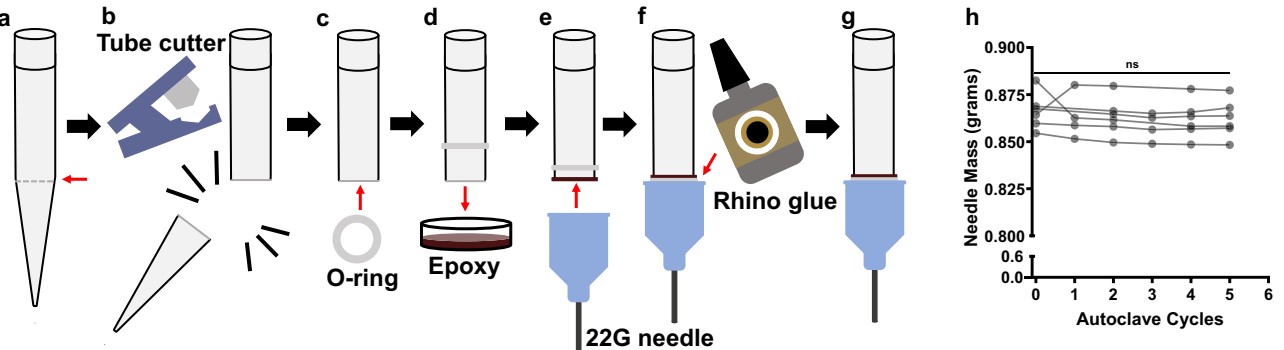

**Fig. 2 | Attachment of 22 G needles to VIAFLO96 tips.** "Long" 300 μL INTEGRA pipette tips were (**a**) marked at the indentation indicated by the red arrow, (**b**) trimmed with a tube cutter, (**c**) followed by addition of an O-ring onto the tip. **d** The female luer connector portion of the needle hub, the average FSS was pipette tips were dipped into epoxy, (**e**) and 22 G needles were secured onto the ends. **f, g** Rhino glue was used to fill any gaps. **h** Summary data for needle weight of $n = 6$ needles after multiple autoclave cycles (One-way ANOVA, *$p < 0.05$).

female luer connector portion of the needle hub, the average FSS was estimated to be 1.57 dyn cm⁻² (0.157 Pa) (Fig. S2b), shown in comparison to the "long" pipette tips (Fig. S2a), determined using the MATLAB calculation to generate Fig. 1.

Although the FSS varies within these regions, we are primarily concerned with maximizing the FSS the cells experience while traveling through the needle, however the FSS that a cell experiences within any position in the modified tips is still within a physiological range[3,20,47]. The maximum shear rate that the 22 G needles produce at 295 μL s⁻¹ is $4.36 \times 10^4$ s⁻¹, with an average of $2.90 \times 10^4$ s⁻¹. These values are also consistent with shear rates present in cancer patients who show symptoms such as inflammation within larger arteries, which can range from $2.00 \times 10^3$ s⁻¹ to $4.00 \times 10^5$ s⁻¹ [24,48–50].

The efficacy of the 22 G needles was first tested by examining PCa cell mechanotransduction under the application of FSS and TRAIL, following the process overview depicted in Fig. 3a–c. This procedure is broadly adaptable to test other cell types, or therapeutics to be coupled with biochemical assays, especially those based on instruments configured to analyze multiwell plates, such as spectrophotometers and flow cytometers.

### Increasing VIAFLO96 shearing cycles enhanced TRAIL-mediated apoptosis in prostate cancer cells

The induction of TRAIL-mediated apoptosis was tested in two different PCa cell lines: LNCaP and PC3, shown in Fig. 4. Both PCa cell lines were exposed to a minimum of 500 shearing cycles ( ~ 20 min). The LNCaP cells were exposed to a maximum of 5000 shearing cycles ( ~ 200 min) since these cells have been previously shown to be more sensitive to high intensity FSS (395 Pa), whereas the PC3 cells were exposed to a maximum of 10,000 shearing cycles ( ~ 400 min) since PC3s have exhibited a level of innate resistance to FSS[3,20,44]. A single shearing cycle represents one "mix" in the VIAFLO96, implying that the cells experience the average FSS of 290 dyn cm⁻² twice during one shearing cycle. During each shearing cycle, the same total volume of cell suspension was aspirated and dispensed to minimize dead volume within the 22 G needle region of the flow domain.

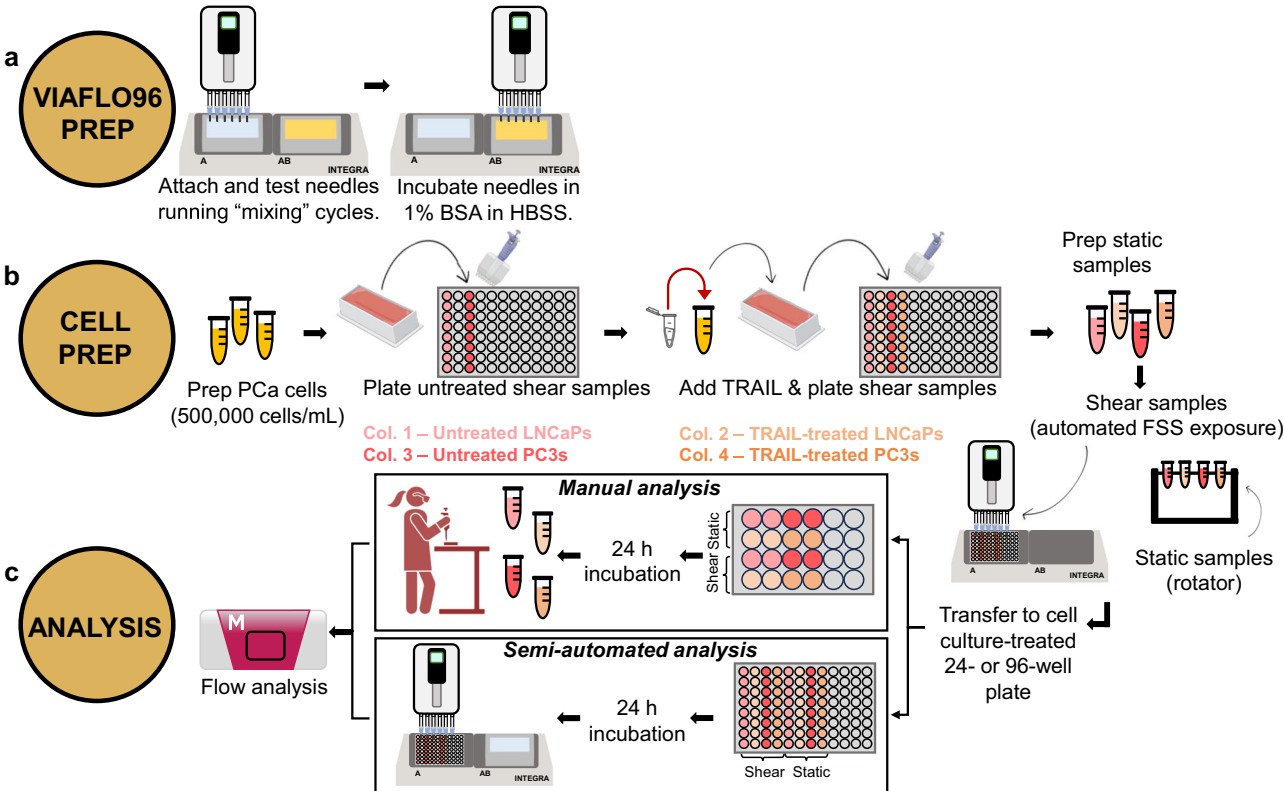

**Fig. 3 | Process overview for VIAFLO96 manual and semi-automated experimentation following TRAIL treatment. a** Needle attachment and 1% BSA incubation arrangement to prepare the VIAFLO96. **b** Procedure to prepare the PCa cells with and without TRAIL for FSS exposure in a 96-well plate. **c** Downstream procedural overview for manual and semi-automated methods for flow cytometry preparation.

The approximate durations for the shearing programs are listed in Table S3, and Table S4 provides an example of the commands used in a VIALINK program to expose cells to 5000 shearing cycles. Cells were also treated with or without 500 ng mL$^{-1}$ TRAIL and static conditions were prepared as controls. The percentage of viable and apoptotic cells was determined using Annexin-V/ propidium iodide (AV/PI) staining in flow cytometry 24 h after FSS exposure since the duration of downstream apoptosis to fully occur can range from a few minutes to hours[28,51–53]. Although the FSS that the cells are exposed to (290 dyn cm$^{-2}$) is moderately higher than the FSS that a cell is thought to experience in venous and arterial circulation, the FSS intensity can be modified by lowering the flow rate (Table S5) in the VIALINK programs to more directly correspond with the venous and arterial circulatory range (0.05–30.0 dyn cm$^{-2}$)[3,20].

For both PCa cell lines, Fig. 4 shows that as the shearing cycles increased in number, apoptosis for the TRAIL-treated conditions increased (Fig. 4b, d). At the maximum FSS exposure, the percentage of viable LNCaP and PC3 cells for the combination-treated group decreased to about 20% (Fig. 4a, c). FSS exposure alone did not cause a significant change in cellular apoptosis. An increase in TRAIL-sensitization consistent with an increase in duration of FSS exposure further supports these results (Fig. 4h), calculated using the viability determined from the AV/PI flow cytometry assay (Fig. 4e–g). TRAIL sensitization is a measure of the extent to which FSS alone is inducing the pro-apoptotic effects of TRAIL[28]. For LNCaP cells, the TRAIL sensitization was 78% at 5000 shearing cycles, and for the PC3 cells, this value reached 69% following exposure to 10,000 shearing cycles. Representative brightfield images of the LNCaP and PC3 cells following exposure to 5000 shearing cycles are shown in Figure S3. Initially, we examined the viability for static conditions placed on the rotator or in wells of the 96-well plate exposed to 0 dyn cm$^{-2}$, and determined that there was insignificant variation between each method. Therefore, all experiments were

conducted with static samples on the rotator to reduce cell settling while the shear samples were exposed to FSS. Previous research in our lab also determined that differences were negligible between samples on a rotator and "0 dyn cm$^{-2}$" static conditions in which cells were loaded into the shear apparatus identical to the FSS conditions, with the device remaining in the "off" state[42]. Following FSS exposure, a change in media color was also visually apparent, suggesting ATP release from the cancer cells under stress, resulting in a change in pH[54]. These results demonstrate that the therapeutic effects of soluble TRAIL are significantly enhanced by exposure to physiological FSS.

## Mechanically-induced mitochondrial depolarization confirms intrinsic apoptosis

TRAIL-mediated apoptosis has been shown to occur through synergistic cross-talk in cancer cells between the extrinsic apoptotic pathway induced by death receptor 4/5 recognition of TRAIL, and the intrinsic apoptotic pathway via FSS activation of Piezo1. This results in successful execution of the caspase cascade[28,52]. The percentage of depolarized mitochondria was detected as a decrease in red fluorescence in the JC-1 flow cytometry assay 24 h after FSS exposure. Figure 5 shows that the degree of mitochondrial depolarization was highest for the FSS + TRAIL-treated groups for both PCa cell lines, and that this percentage increased as the shearing cycles increased.

These results are consistent with the enhanced apoptosis evident in the FSS + TRAIL-treated samples (Fig. 4b, d), suggesting that the intrinsic apoptosis pathway in PCa cells is activated due to mechanical activation of Piezo1 via the applied FSS[28]. After 5000 shearing cycles, 76% of LNCaP cells also treated with TRAIL underwent mitochondrial depolarization, 20% higher than the LNCaPs in the static condition treated with TRAIL only (Fig. 5a, c). An elevated percentage of cells showing mitochondrial depolarization for the static+TRAIL LNCaP cells (58%) is likely because the

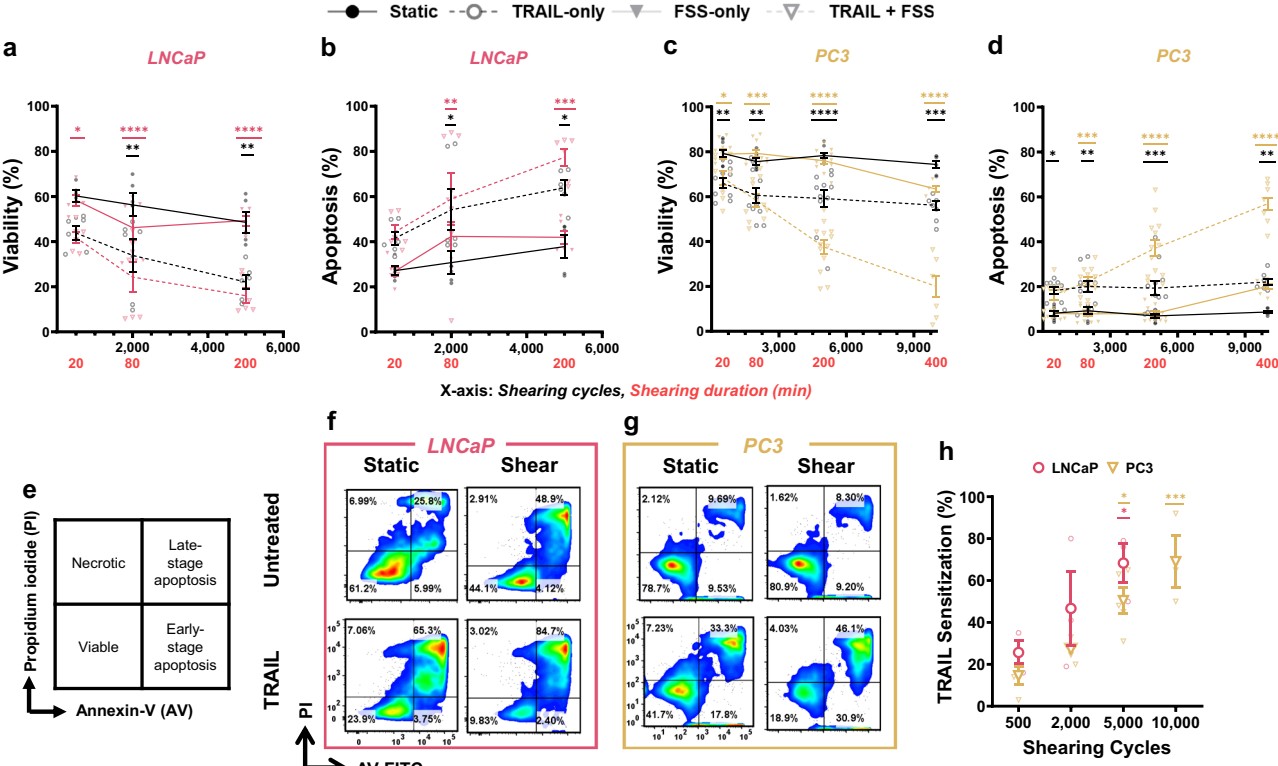

**Fig. 4 | Fluid shear stress induces TRAIL sensitization in PCa cells.** Summary data for the percentage of (**a**) viable and (**b**) apoptotic LNCaP cells and (**c**) viable and (**d**) apoptotic PC3 cells treated with TRAIL, following increasing fluid shear stress exposure using 22 G needles in the VIAFLO96 instrument ($n = 3$–$5$ independent experiments; two-way ANOVA test, (Supplementary Data 1); colored significance stars show the comparison between TRAIL + FSS and static conditions at each shearing duration; black significance stars compare the TRAIL-only condition to the static condition at each shearing duration). **e** Graphical interpretation for AV/PI flow cytometry staining. Representative AV/PI flow cytometry plots for (**f**) LNCaP and (**g**) PC3 cells exposed to 5000 shearing cycles. **h** TRAIL sensitization calculated using Eqs. 4 and 5, from the viability measured with the AV/PI assay ($n = 4$ independent experiments; unpaired $t$-test comparing the TRAIL + FSS to the TRAIL-only condition). *$p < 0.05$, **$p < 0.01$, ***$p < 0.005$, ****$p < 0.0001$. Error bars represent ± SEM.

LNCaP cells were derived from the lymphatic system, where they experienced much lower levels of FSS in this region. When the static samples are on the rotator, they still experience very minimal exposure to FSS ($< 0.05$ dyn cm$^{-2}$), and increased durations of this constant motion is likely affecting the LNCaP cell response, potentially causing some mechanoresponse[3,42,55,56]. At 10,000 shearing cycles, the percentage of PC3 cells also treated with TRAIL showed a 4-fold higher percentage of depolarized mitochondria than all other treatment groups (Fig. 5b, d). Furthermore, FSS exposure alone did not significantly affect mitochondrial depolarization. Overall, this effect was more pronounced in the LNCaP cells (Fig. 5a, c), consistent with results that have shown LNCaP cells to be more sensitive to FSS-induced cell death[3].

### Manual and semi-automated downstream staining methods produce consistent results

Semi-automated AV/PI and JC-1 staining programs were developed using VIALINK to allow the researcher to take advantage of the multiplex capabilities of the VIAFLO96. To confirm the accuracy and efficiency of these programs, the PC3 cells were exposed to 5000 shearing cycles and were plated overnight in either a 24- or 96-well plate. Analysis was performed 24 h later using either a manual or semi-automated staining protocol (Fig. 6). The materials required for the downstream semi-automated method are listed in Supplementary Data 1.

Minor differences produced by the two staining methods for AV/PI viability (Fig. 6a, b, d) and mitochondrial depolarization (Fig. 6c) are noted, but do not reach statistical significance. Representative JC-1 flow cytometry plots are shown in Figure S4. TRAIL sensitization trends were comparable for the manual and semi-automated staining methods (Fig. 6e). This finding

demonstrates the potential to use the VIAFLO96 to scale-up flow cytometry experiments in a multiplex manner, either following exposure to FSS as demonstrated here, or to test therapeutic responses under static conditions. The VIALINK semi-automated staining programs included with this article have been scripted to process multiple cell lines at once based on the experimental configuration in Fig. 3, and these programs can be easily modified with the user-friendly interface of the VIAFLO96 device and VIALINK software to process each cell line individually (as shown in Fig. 6), and various numbers of samples simultaneously depending on the needs of the experiment.

### Mechanosensitive ion channel activation contributes to TRAIL-mediated apoptosis

To confirm MSC activation by FSS (i.e., Piezo1), PC3 cells were pretreated with 10 μM of GsMTx-4, a pharmacological inhibitor of cationic MSCs, followed by TRAIL treatment and exposure to 5000 shearing cycles[16,57,58]. In Fig. 7, PC3 cells treated with GsMTx-4 and FSS + TRAIL showed reduced TRAIL-induced apoptosis (Fig. 7a, c, d) and mitochondrial depolarization (Fig. 7b, e) compared to the FSS + TRAIL-treated conditions without the MSC inhibitor.

The degree of TRAIL sensitization was determined by considering the GsMTx-4-treated samples exposed to FSS[28,52]. Pre-treatment of PC3 cells with GsMTx-4 reduced TRAIL sensitization by about 1.5-fold for the shear groups, and remained nearly unchanged for the static conditions. However, since there is no significant variation observed for the TRAIL sensitization between the shear conditions (Fig. 7f), it is possible that additional MSCs not fully inhibited by GsMTx-4 are contributing to these effects[59–61]. The use of GsMTx-4 to inhibit Piezo1 is justified by numerous studies that have yielded

**Fig. 5 | TRAIL-mediated apoptosis occurs through mitochondrial depolarization.** Summary data for the percentage of depolarized mitochondria for (**a**) LNCaP and (**b**) PC3 cells exposed to increasing shearing cycles using the 22 G needles with the VIAFLO96 instrument combined with TRAIL treatment ($n$ = 3–4 independent experiments; two-way ANOVA test, (Supplementary Data 1); colored significance stars show the comparison between TRAIL + FSS and static conditions at each shearing duration; black significance stars compare the TRAIL-only condition to the static condition at each shearing duration). Representative JC-1 flow cytometry plots for (**c**) LNCaP and (**d**) PC3 cells after 5000 shearing cycles. *$p < 0.05$, **$p < 0.01$, ***$p < 0.005$, ****$p < 0.0001$. Error bars represent ± SEM.

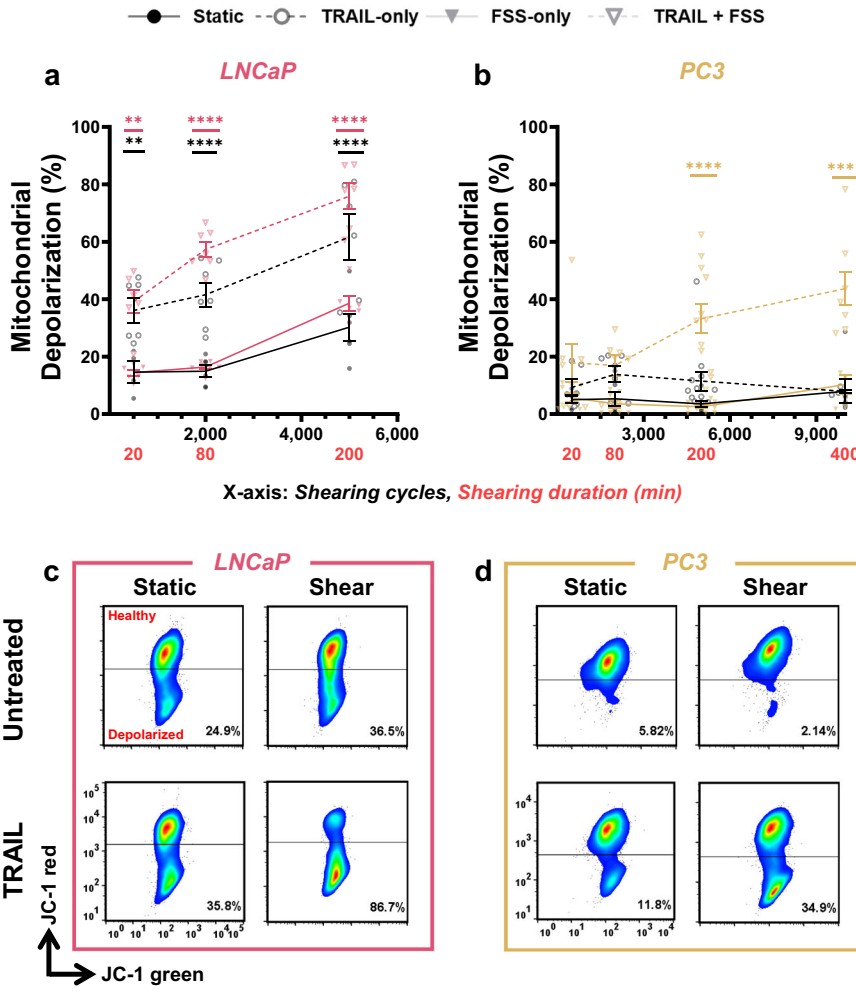

comparable results amongst pre-treatment of cells with GsMTx-4 and Piezo1 silencing or knockdown in cell lines[27,28,62–64]. This finding suggests that MSC activation results from the application of FSS to induce synergistic cancer cell apoptosis in both non-metastatic and invasive PCa cells, as hypothesized.

### Confirmed induction of intrinsic apoptosis within small bore needles

PC3 cells were also exposed to FSS using the unmodified "standard" INTEGRA pipette tips, to compare and confirm that the increased FSS within the 22 G needles produced increased mechanoactivation. At 2000 shearing cycles, the 22 G needles induced a greater degree of apoptosis for the TRAIL-treated PC3 cells (Fig. 8a–c) compared to the unmodified pipette tips. The TRAIL concentration was kept constant at 500 ng mL$^{-1}$. Increased TRAIL sensitization for the PC3 cells sheared with the 22 G needle system compared to "standard" tips further confirms these findings (Fig. 8d). Furthermore, the 22 G needles induced higher mitochondrial depolarization for the FSS + TRAIL conditions compared to the unmodified tips (Fig. 8e, f).

The TRAIL-mediated apoptosis that PC3 cells demonstrated after being exposed to 2000 shearing cycles using the 22 G needle system in Fig. 8 is consistent with those obtained in Fig. 4c, d, f. These results lead us to conclude that the extended, higher-intensity FSS that cells experience in the 22 G needles more successfully induced the pro-apoptotic effects of TRAIL under FSS conditions using the VIAFLO96 compared to the unmodified, "standard" INTEGRA pipette tips.

### Shear stress enhances murine dendritic cell differentiation ex vivo

To expand these methodologies to study immune cell mechanotransduction and further test the efficacy of the VIAFLO96 apparatus to mimic biologically relevant FSS in vitro, we developed a semi-automated protocol to fix, permeabilize, and intracellularly stain DCs for flow cytometry. The process overview is shown in Fig. 9a–c. For these experiments, we used primary, murine BMDCs.

A procedural overview for the BMDC isolation is shown in Fig. 10a. Briefly, the BMDCs were isolated on day 0 and treated with granulocyte macrophage-colony stimulating factor on day 0 and day 3. On day 6 the BMDCs were differentiated and ready to be used for experiments[65]. We exposed the DCs to 5000 shearing cycles using the 22 G needles fitted to the VIAFLO96 system[65]. The downstream, intracellular flow cytometry staining was completed using multiplex, semi-automated programs developed in VIALINK, with the materials listed in Supplementary Data 1. Previous experiments, including some within our lab, have examined the effects of physiological FSS on primary, murine BMDCs, and these studies have previously confirmed that DC exposure to FSS alone does not negatively impact DC viability[42,43].

Following FSS exposure, BMDC differentiation is enhanced (Fig. 10b, c), denoted by increased CD11c expression compared to the static condition[42,66,67]. Additionally, expression of co-stimulatory molecules MHC-II and CD80 following FSS exposure were significantly higher than for cells under static conditions, which is critical to elicit adaptive immune responses[68,69]. The MHC-II expression after FSS exposure was 2.4- and

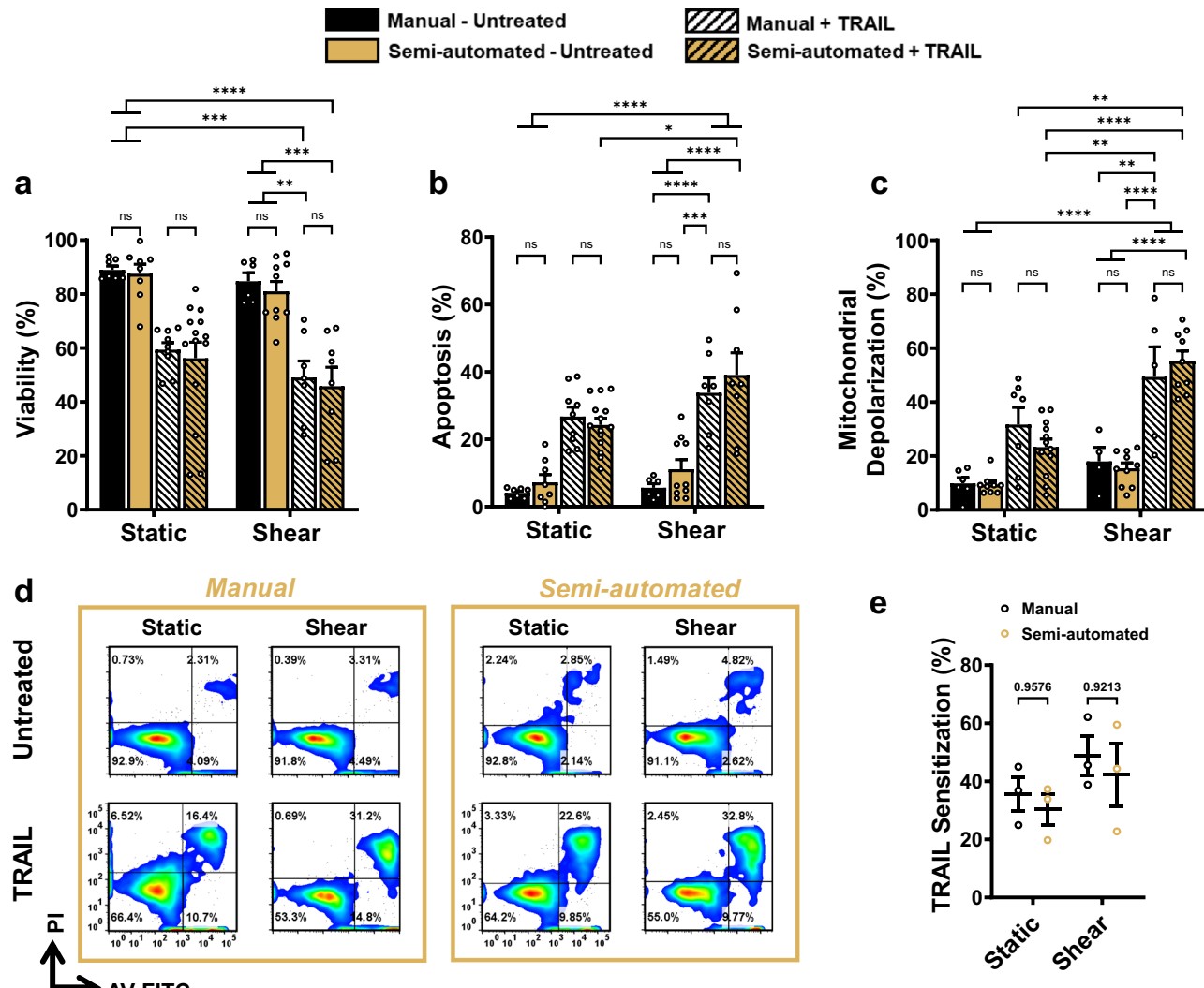

**Fig. 6 | Manual and semi-automated flow cytometry staining produce consistent results.** PC3 cells exposed to 5000 shearing cycles using the VIAFLO96 equipped with 22 G needles with TRAIL treatment. Summary data for the percentage of (**a**) viable and (**b**) apoptotic PC3 cells. **c** Summary data for the percentage of depolarized mitochondria. Samples are stained manually or using the semi-automated program produced using VIALINK. **d** Representative AV/PI flow cytometry plots. **e** TRAIL sensitization from the viability generated in (**a**) were calculated using Eqs. 4 and 5. ($n = 4$ independent experiments (**a**, **b**, **e**), $n = 3$ independent experiments (**c**); two-way ANOVA test (**a–c**, **e**), (Supplementary Data 1)). *$p < 0.05$, **$p < 0.01$, ***$p < 0.005$, ****$p < 0.0001$. Error bars represent ± SEM.

1.3-fold higher than previous reports that exposed BMDCs to IL-4 as a stimulant[69]. Significantly increased phosphorylated-nuclear factor-kappa B was also observed. This is critical since phosphorylated-nuclear factor-kappa B modulates DC maturation and function, such as the regulation of inflammatory responses by driving T cell priming through cytokine release[41,42]. Previous research in our laboratory has assessed the use of potent stimulators, including lipopolysaccharide to activate DCs. Exposure to 1 h of low-intensity FSS determined that DC proliferation was significantly increased and Rac1 expression was enhanced compared to the static condition with or without lipopolysaccharide[42]. These trends are indicative of enhanced DC activation[42]. Furthermore, BMDCs treated with lipopolysaccharide yielded MHC-II expression lower than the results presented here for the FSS-treated group (~ 10%)[42,68]. Therefore, we did not compare the results in Fig. 10 to treatment with a chemical activator since we have focused on examining the performance of our modified tips integrated with our semi-automated flow preparation protocols to provide easily adoptable methods.

Studying DC response to mechanical stimuli is critical since DCs may encounter mechanical strain while residing on surfaces such as arterial walls in the bloodstream. Exposure to these forces in vitro have shown similar trends in which MHC-II, CD86 and CD80 expression increased following exposure to mechanical strain[70]. These observations demonstrate the potential to activate DCs ex vivo to enhance immune cell response in cancer therapies.

## Discussion
The results presented here are consistent with existing studies that have used previously established, known methods to delineate the effects of low intensity FSS (2 dyn cm$^{-2}$) exposure on TRAIL-sensitization of cancer cells[28]. The protocols here recreate this phenomenon in a straightforward, high-throughput method to effectively apply physiologically relevant FSS to cells. This was achieved using a multiplex approach in which the effects of physiological forces on different cell types and conditions can be investigated within the same multiwell plate. Understanding the response of cancer cells under constant stress while exposed to TRAIL is imperative for engineering solutions to overcome the poor half-life of soluble TRAIL in circulation, while eliminating the need for complex delivery vehicles[71,72]. Studies that have investigated the exploitation of physiological FSS to induce TRAIL sensitization have only explored lower FSS intensities[28,32,73].

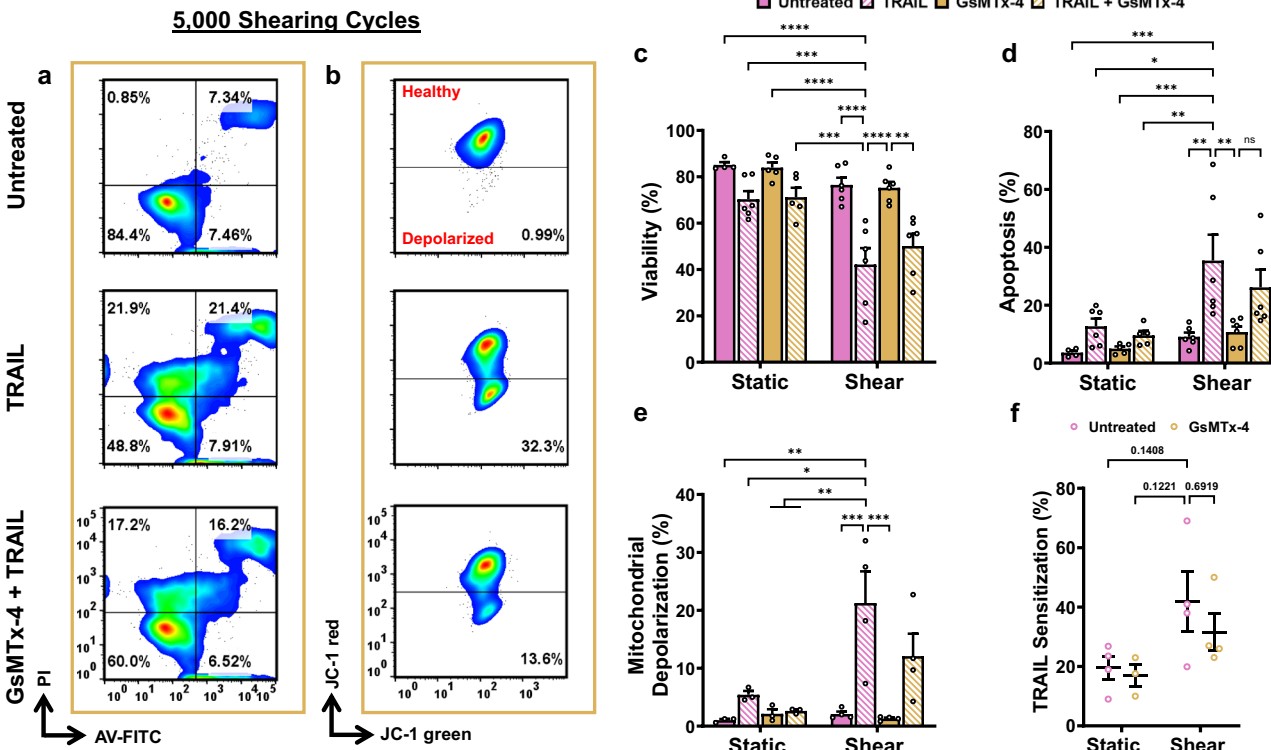

**Fig. 7 | Mechanosensitive ion channel inhibition reduces TRAIL sensitization.** Representative (**a**) AV/PI and (**b**) JC-1 flow cytometry plots. Summary data for the (**c**) viability, (**d**) apoptosis, (**e**) depolarized mitochondria and (**f**) TRAIL sensitization of PC3 cells exposed to 5000 shearing cycles using 22 G needles equipped with the VIAFLO96 system with TRAIL and GsMTx-4 treatment, calculated using Eqs. 6 and 7 (n = 4 independent experiments (**c**, **d**, **f**), n = 3 independent experiments (**e**); two-way ANOVA test). *$p < 0.05$, **$p < 0.01$, ***$p < 0.005$, ****$p < 0.0001$. Error bars represent ± SEM.

Here, we examine the pro-apoptotic effects of TRAIL, mechanically induced by FSS over an interval of almost seven hours of varied FSS durations (Figs. 4, 5). A nearly eight-fold increase in TRAIL sensitization for PCa cell lines was observed as a function of shearing cycle (Fig. 4h). The increased mitochondrial depolarization for both LNCaP and PC3 cells (Fig. 5) for the TRAIL + FSS conditions directly corresponded with increasing apoptosis (Fig. 4), showing a monotonic dependence on shearing cycle. These observations are consistent with previous work by Hope et al. and confirmed that the FSS-induced TRAIL-pro-apoptotic effects resulted from the intrinsic mechanism[28]. Inhibition of Piezo1 prior to and during FSS exposure for PC3 cells demonstrated less cell death for the TRAIL + FSS condition, compared to samples not treated with GsMTx-4 (Fig. 7). Research has shown that GsMTx-4 is less effective or unable to inhibit TRPV4 channels in various cell types, while successfully targeting Piezo1 for inhibition[16,28,52,74]. Previous studies from our group have also investigated the efficacy of treating cells with Yoda1 to chemically activate Piezo1, combined with TRAIL treatment, which have indicated that FSS is comparably effective to synergistically induce TRAIL sensitization[28,52]. This finding provides further evidence that the FSS produced within small-bore needles in combination with the VIAFLO96 is reliable and effective from one experiment to the next.

Further, comparing the results for the AV/PI manual and semi-automated flow cytometry staining protocols (Fig. 6) confirmed that consistent results can be acquired using either method. This consistency permits flexibility to the researcher to instill the manual staining methods when only a few samples will be studied, while also providing the option to perform the semi-automated staining programs established using VIALINK to process a larger number of samples. The cellular staining programs produced in VIALINK are semi-automated since the researcher is required to manually transfer the 96-well plate to a centrifuge to pellet the cells for washing steps.

However, this is only a minor inconvenience for larger-scale experiments, since this entails less human intervention than what would be required if the operator was to manually stain and prepare all the samples for analysis in microcentrifuge tubes with a table-top aspirator, while also reducing the opportunity for human error.

We explored the versatility of these protocols to study FSS-induced activation of primary BMDCs through developing intracellular staining programs. This exemplifies the multiplex capabilities of the VIAFLO96 by providing a smooth translation to downstream analysis to simultaneously stain for several activation markers at once. Exposure to 290 dyn cm$^{-2}$ using the VIAFLO96, compared to 5 dyn cm$^{-2}$ using cone-and-plate viscometers, promoted a more significant increase in expression of co-stimulatory molecule CD80 (Fig. 10b, c)[42]. The interaction of CD80 with CD28 on T cells is necessary for CD4+ T cell activation, thus it would be interesting in future work to expose co-cultures of DCs with naïve T cells to FSS[75–77]. Enhanced phosphorylated-nuclear factor-kappa B expression can be indicative of numerous intracellular biomechanical responses by DCs following FSS exposure, such as activation of Piezo1 or other MSCs[42]. This may also be further tested by treating the DCs with GsMTx-4 prior to FSS exposure. These results provide evidence that FSS applied to primary DCs ex vivo can be translated to fabricate DC-based immunotherapies, potentially making them more accessible to patients.

A limitation to this method is the highest magnitude of shear stress that can be administered to the cells. Increasing the maximum shear stress beyond what is reported for this device operating with 22 G needles (290 dyn cm$^{-2}$) is unattainable in its current form due to the pressure changes associated with the operation of the VIAFLO96 pipetting head. These operating parameters enable the application of these protocols to examine the lower range of physiologically-relevant FSS, as opposed to high-intensity FSS briefly experienced at locations of turbulent flow in the human

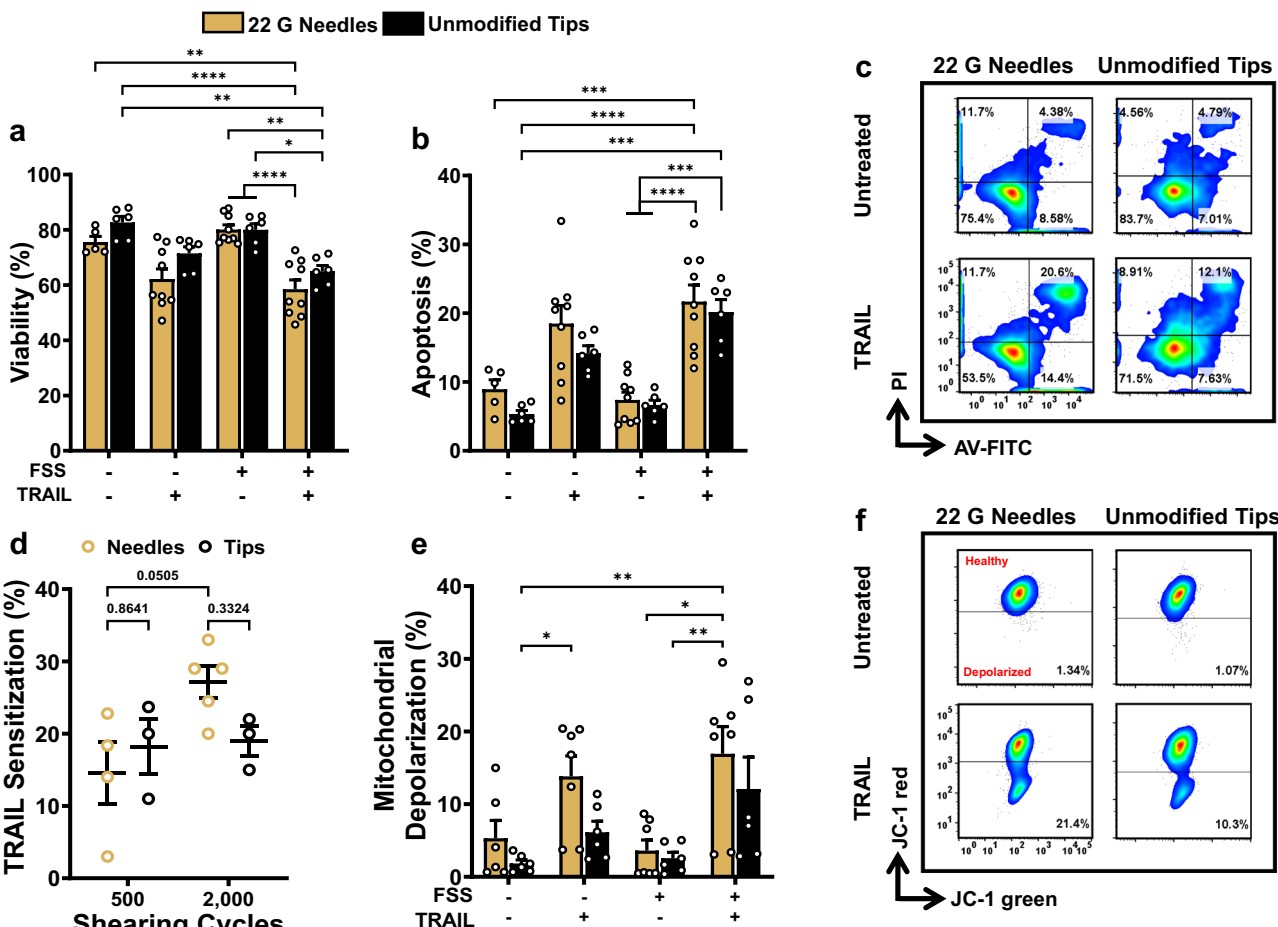

**Fig. 8 | 22 G needles induce higher pro-apoptotic effects of TRAIL compared to unmodified INTEGRA pipette tips.** Summary data for the (**a**) viability and (**b**) apoptosis of PC3 cells, and (**c**) representative AV/PI flow cytometry plots. **d** TRAIL sensitization calculated using Eq. 5 from the viability after fluid shear stress exposure. **e** Summary data for the percentage of depolarized mitochondria and (**f**) representative flow cytometry plots. Data shown for PC3 cells exposed to 2000 shearing cycles using the VIAFLO96 ($n$ = 3–5 independent experiments; two-way ANOVA test (**a, b, d, e**)). *$p < 0.05$, **$p < 0.01$, ***$p < 0.005$, ****$p < 0.0001$. Error bars represent ± SEM.

body[19–21]. The VIALINK shearing programs may also be used with INTE-GRA pipette tips if the elevated FSS using the 22 G needles is not desired. Research using devices such as parallel-plate chambers have also explored the effects of FSS above venous and arterial range, as well[34,78].

These results provide evidence that this method of applying physiologically relevant FSS can reproduce findings achieved using previous methods to apply FSS in vitro and ex vivo[3,28,42,43]. The ability to combine this wide parameter range directly with versatile analytical assays such as flow cytometry can improve laboratory workflow in mechanobiology studies, such as examining cellular response to single and combination therapies of existing drugs and those in development. The VIALINK programs may also be tailored and optimized to an individual lab's purposes to prepare samples for flow cytometry, based on the antibodies required, targets of interest, or desire to sort cells after FSS exposure based on mechanoadaptive responses, while permitting flexibility to use a manual or semi-automated downstream approach. These protocols can be adapted to other downstream quantitative assays that require a multiplex approach including ELISAs, Seahorse, luciferase reporter assays, and a considerable number of other techniques.

This technology is particularly useful for researchers in need of a semi-automated method to test the response of tumor cells, immune cells or patient-isolated CTCs to various types or doses of therapeutics in combination, either with or without the application of physiological FSS. Implementing pre-established methods to expose cells to physiological FSS

in vitro such as cone-and-plate viscometers or the syringe pump requires manual manipulation and handling following FSS treatment. Further, these devices are generally single-chamber systems, unlike the VIAFLO96 used in conjunction with multiwell plates. The protocols herein can also be adapted to the VIAFLO384 device, which operates with a 384-channel pipetting head to increase the sample number and type and enhance the multiplex capabilities beyond those presented here. The intracellular staining programs can be translated to use with the PCa studies to build off those explored here, to probe additional markers involved in mechanotransduction-mediated apoptosis and mechanoresponses to FSS. Protocols at the maximum FSS (290 dyn cm$^{-2}$) may also be adapted to study the effects of atherosclerosis in cancer, since patients with diseased arteries express shear rates up to 40,000 s$^{-1}$, whereas our modified system provides a maximum shear rate of 43,569 s$^{-1}$[48,79]. Continuing and advancing studies of how we can enhance the response and activation of immune cells to physiological forces is imperative in cancer since CTCs can evade immune surveillance. In future work, it would be interesting to explore how multiplex FSS affects co-cultures of DCs with PCa cells since these results provide a foundation to study the direct biophysical interaction between immune cells and cancer cells during circulation through the body, or to study the ex vivo activation of splenocytes derived from orthotopic cancer models. The protocols established herein provide an alternative and cost-effective means for scaling-up experiments, that are not uniquely limited to the field of cancer research.

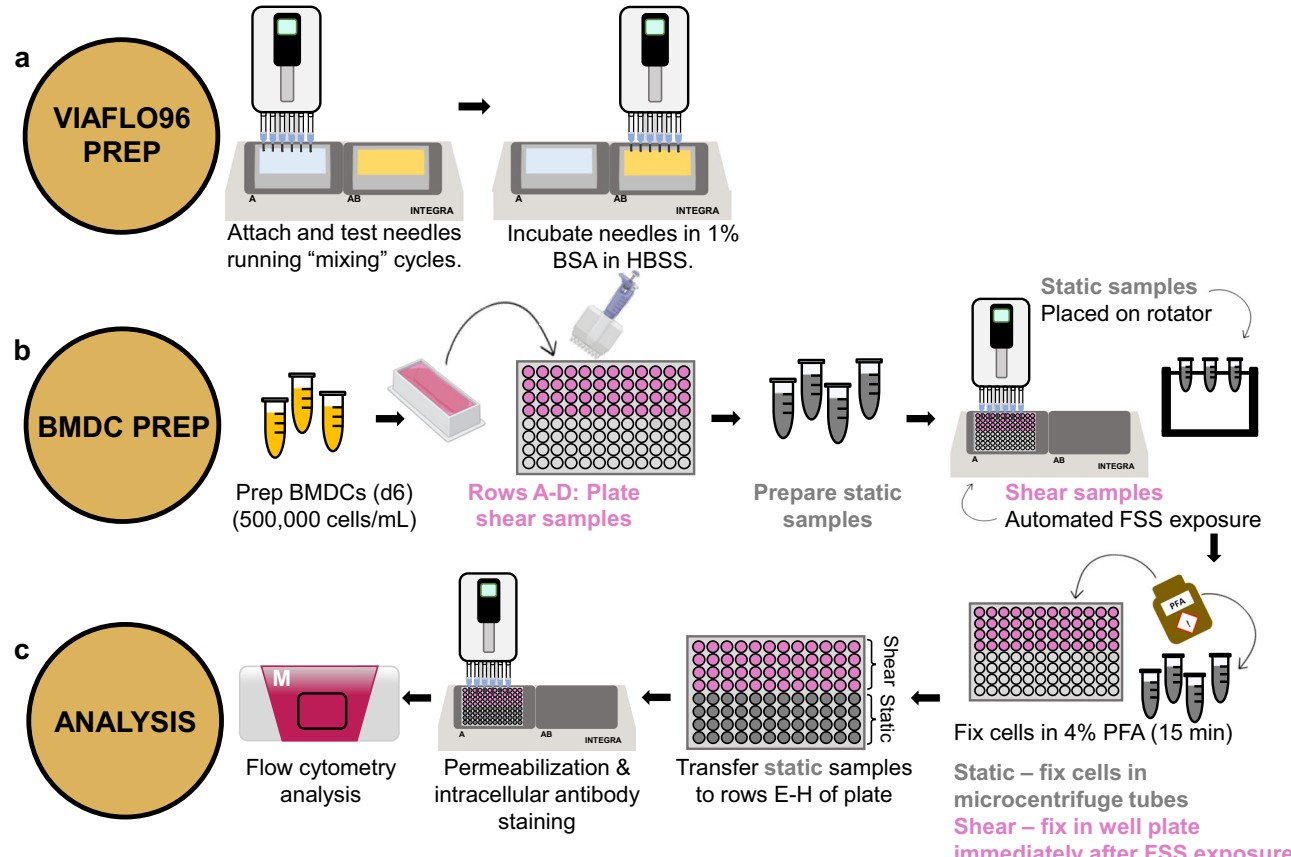

**Fig. 9 | Process overview using the VIAFLO96 to test the effects of fluid shear stress on primary immune cells. a** 22 G needle attachment and 1% BSA incubation arrangement to prepare the VIAFLO96. **b** Procedure to prepare the static and shear samples of BMDC cells. **c** Downstream semi-automated overview for intracellular flow cytometry preparation.

## Methods

### 22 G needle assembly

The "long" 300 μL pipette tips (INTEGRA Biosciences, New Hampshire, USA) that are compatible with the VIAFLO96 (INTEGRA Biosciences) were cut with a pipe cutter (Allen Technologies, USA). A silicone O-ring (Uxcell, Hong Kong, China) was placed onto the tip, and the tip was dipped into a fresh MarineWeld epoxy (JB-Weld, USA) solution (without submerging the O-ring). The O-rings have a 5 mm outer diameter, 1.5 mm inner diameter, and 1.5 mm width. The 22 G needle hub (Cellink, California, USA) was latched onto the end of the tip with epoxy, and the O-ring was lowered down to secure the cavity between the 22 G needle hub to the pipette tip. A small layer of Rhino Glue (Rhino Glue, California, USA) was applied right above the O-ring to eliminate any remaining air pockets. The resulting needle assemblies were allowed to dry overnight and autoclaved before use.

The FSS that a cell experiences in the 22 G needle was estimated using Poiseuille's equation:

$$\tau_{max} = \frac{4Q\mu}{\pi R^3} \quad (1)$$

Where $\tau$ is the wall shear stress (dyn cm$^{-2}$), $Q$ is the flow rate (cm$^3$ s$^{-1}$) which is 18 mL min$^{-1}$, $\mu$ is the viscosity of the RPMI medium, which is assumed to be water at room temperature and standard pressure (0.01 dyn s cm$^{-2}$), and $R$ is the inner radius of the 22 G needles (2.05 × 10$^{-2}$ cm). The maximum FSS is located at the wall of the conduit since the local FSS varies linearly with radial position. This leads to an area-averaged FSS equal to two-thirds of the maximum (290 dyn cm$^{-2}$ or 29 Pa). A single pulse through the needles has a residence time of 5.69 ms. The cancer cells were exposed to

500–10,000 shearing cycles, corresponding to a runtime of 20 ~ 400 min. Following FSS exposure, the suspended cells were plated in a 24-well (CELLTREAT, Pepperell, MA, USA) or 96-well plate (Corning, Manassas, VA, USA).

The minimum FSS in a conduit may be estimated using the following equation:

$$\tau_{min} = \tau_{max}\left(\frac{r}{R}\right) \quad (2)$$

where $r$ is the radius of the cell, estimated in this case to be 9.31 μm[20,46]. The Reynolds number for this flow was calculated to confirm laminar flow and justify the use of Poiseuille's equation, and is defined as follows:

$$Re = \frac{\rho v D}{\mu} \quad (3)$$

where $\rho$ is the buffer density, assumed to resemble water at standard room temperature and pressure (0.998 g cm$^{-3}$), $v$ is the flow velocity, $D$ is the inner needle diameter, and $\mu$ is the buffer viscosity. The Reynolds number at a flow rate of 18 mL min$^{-1}$ is 914. This value is below 2200, the threshold for laminar flow, implying that Poiseuille's equation is appropriate to predict the average FSS.

### Cell culture

Prostate adenocarcinoma cell lines PC3 (ATCC #CRL-1435) and LNCaP (ATCC #CRL-1740) were purchased from American Type Culture Collection (Manassas, VA, USA). LNCaP and PC3 cells were cultured in RPMI 1640 cell culture medium (Gibco, Paisley, UK), both supplemented with

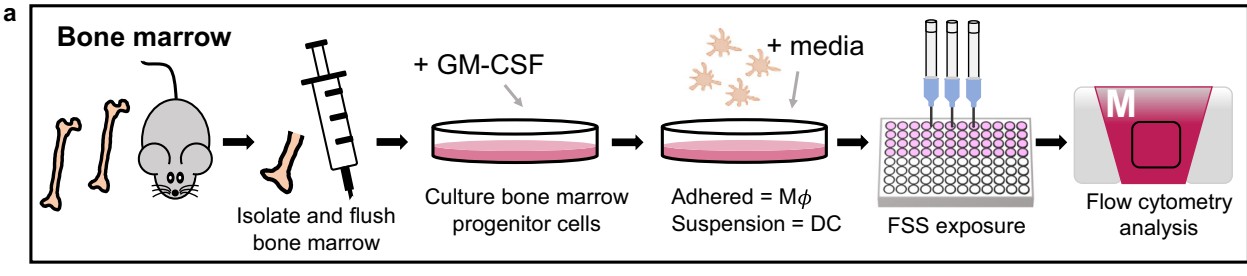

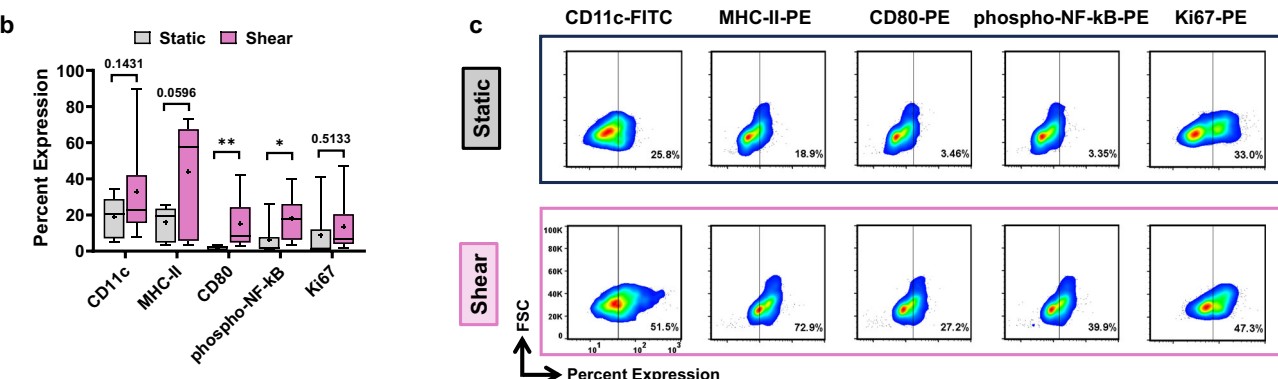

**Fig. 10 | Enhanced differentiation of murine BMDCs. a** Procedural overview for the isolation and exposure of BMDCs to fluid shear stress. **b** Summary data and (**c**) representative flow cytometry plots for the percent expression of various BMDC differentiation and co-stimulatory molecules following 5000 shearing cycles using 22 G needles with the VIAFLO96 system (*n* = 3 independent experiments; box and whisker plot whiskers extend from min to max, the box extends from the 25th to 75th percentiles, the horizontal line indicates the median and the mean is plotted as (+); unpaired *t*-test comparing the static to the shear condition for each marker examined). GM-CSF granulocyte macrophage-colony stimulating factor, phosphor-NF-kB phosphorylated-nuclear factor-kappa B. *$p < 0.05$, **$p < 0.01$.

10% (v/v) fetal bovine serum (Gibco, Paisley, UK), 1% (v/v) PenStrep (Gibco, Grand Island, NY, USA), and 10 mM HEPES (v/v) (Gibco, Carlsbad, CA, USA). LNCaP media was also supplemented with 1 mM sodium pyruvate (Gibco). Cells were incubated under humidified conditions at 37℃ and 5% $CO_2$, and not allowed to exceed 90% confluence. Cells are regularly mycoplasma tested within the lab.

**Preparation of cancer cells for fluid shear stress experiments**
LNCaP and PC3 cells were washed in HBSS free of $Ca^{2+}$ and $Mg^{2+}$ (Corning, Manassas, VA, USA), followed by treatment with 0.25% trypsin-EDTA (Gibco) for 5–6 min at 37 ℃. Cells were centrifuged at 300 x *g* to remove dissociation agent and resuspended in complete media at a concentration of $5 \times 10^5$ cells mL$^{-1}$ prior to evoking fluid shear stress studies. Cells were exposed to FSS in low-adhesive, round-bottom 96-well plates (Corning).

For studies involving TRAIL, cells were treated with 500 ng mL$^{-1}$ recombinant human TRAIL (PeproTech, Rocky Hill, NJ, USA) immediately preceding FSS exposure. For GsMTx-4 (Alomone Labs, Jerusalem, Israel) experiments, cells were pretreated with 10 µM of the inhibitor, and incubated for 1 h on a rotator prior to fluid shear stress exposure.

**Bone marrow-derived dendritic cell isolation**
In vivo studies were approved by the Vanderbilt IACUC, Protocol #M1700009-02. All methods that used animals or animal-derived samples were carried out in accordance with relevant regulations and guidelines. Mice were monitored by staff from the Division of Animal Care (DAC) at Vanderbilt University. The BMDC isolation followed the protocol by Madaan et al., in which femurs were isolated from healthy, female, eight-week-old BALB/c mice purchased from Jackson Laboratory (Bar Harbor, Maine, USA)[65]. Bone marrow was flushed using HBSS with $Ca^{2+}$ and $Mg^{2+}$ and a 29 G syringe. On day 0, $1 \times 10^6$ bone marrow cells were plated in a single dish and treated with 20 ng mL$^{-1}$ granulocyte macrophage-colony stimulating factor (R&D Systems, Minneapolis, MN, USA) in complete RPMI supplemented with 10% (v/v) fetal bovine serum and 1% (v/v) PenStrep. Cells were incubated under humidified conditions at 37 ℃ and

5% $CO_2$. On day 3, 10 mL of fresh complete RPMI and 20 ng mL$^{-1}$ of granulocyte macrophage-colony stimulating factor was added to the dish. On day 6, macrophages remained adhered to the dish and DCs were in suspension. DCs were then collected and used for experimentation.

For FSS experiments in the VIAFLO96 system, the BMDCs were lifted by gently washing and aspirating the media from the dishes. The cells were washed one time in HBSS with $Ca^{2+}$ and $Mg^{2+}$ by centrifugation at 300 x *g*, and resuspended in complete RPMI at a concentration of 5 x 10$^5$ cells mL$^{-1}$ prior to FSS exposure.

**VIAFLO96 set-up and plating of cells for fluid shear stress exposure**
While preparing cells for experimentation, 22 G needles were manually attached to the VIAFLO96 pipetting head. A 150 mL reservoir (INTEGRA Biosciences) was filled with milli-Q filtered water, with sufficient liquid volume to allow for 2-3 "Pipet/Mix" cycles on the VIAFLO96 to confirm proper operation of the needles (no air pockets, etc.). Once the desired number of needles were loaded onto the device, a 150 mL reservoir was filled with 20–30 mL of 1% (w/v) bovine serum albumin (BSA) (Sigma Aldrich, St. Louis, MO, USA). 200 µL of 1% BSA was pre-incubated into the needles by using the "Pipette" command to aspirate the solution. The tips of the needles were left submerged in solution during this time to prevent clogging.

**Cancer cell experiments.** The following plate set-up and corresponding procedure demonstrates one example of how the well plate could be set up for experimentation (based on Fig. 3): A low-adhesive 96-well plate (Corning) was loaded with 200 µL of cell suspension per well. One column contained untreated LNCaPs and another column was filled with untreated PC3s. Individual columns for LNCaP and PC3 cells treated with TRAIL were also loaded. Static untreated and TRAIL conditions were also prepared in 1.5 mL tubes (Corning).

**BMDC experiments.** A low-adhesive 96-well plate was loaded with 200 µL of cell suspension per well without adding additional granulocyte

macrophage-colony stimulating factor, following the representative plate format based on Fig. 9. Static conditions were also prepared in 1.5 mL tubes to be placed on the rotator.

## Fluid shear stress exposure of cells using the VIAFLO96
Once the 96-well plate was loaded with samples, the needles were set to dispense the 1% BSA by pressing "RUN" on the VIAFLO96 interface. The 96-well plate was placed on Position A of the VIAFLO96 stage (INTEGRA Biosciences), and the desired custom shearing program was started by pressing "RUN" (from "Custom Programs"). A small piece of foil was taped around the area exposed between the pipetting head and plate on the stage to reduce volume loss due to evaporation, and the static samples were placed on a rotator corresponding to the same duration as the shear exposure being investigated. The shear exposure ranged from 500 to 10,000 shearing cycles (runtime of 20 ~ 400 min).

**Cancer cell experiments**. After shear exposure, the samples were either transferred to a culture-treated 24- (manual analysis) or 96-well plate (semi-automated analysis) for 24 h incubation under humidified conditions at 37 °C and 5% $CO_2$. For the 24-well plate, samples were transferred using "96 TO 24 COL1-4.xml" and "96 to 24 COL5-8.xml" VIALINK programs. For the 96-well plate, samples were transferred using the VIALINK program "96 TO 96.xml". These VIALINK programs are provided as supplemental files and can be easily modified depending on the number of samples or conditions being tested. After incubation, samples were analyzed for cell death using the AV/PI assay or mitochondrial depolarization using the JC1 assay via flow cytometry.

24 h after incubation in a 24-well plate, an Olympus IX81 inverted microscope (Olympus) was used to acquire representative 20X brightfield images of the LNCaP and PC3 cells exposed to 5000 shearing cycles.

**BMDC experiments**. The BMDCs were exposed to 5000 shearing cycles (200 min). Following shear exposure, a 150 mL reservoir was filled with 20 mL of a 16% paraformaldehyde (Electron Microscopy Sciences, Hatfield, PA, USA) aqueous solution in HBSS with $Ca^{2+}$ and $Mg^{2+}$. Once the VIALINK shearing program was complete, the "Pipet/Mix" command on the VIAFLO96 interface was used to fix each well containing BMDCs in suspension with paraformaldehyde at a final concentration of 4%. The static samples were fixed in the microcentrifuge tubes using paraformaldehyde diluted to a final concentration of 4%. Following a 15 min fixation period, 200 μL of the static samples were manually pipetted into rows E – H of the well-plate.

The 22 G needles were rinsed with 70% ethanol, followed by HBSS free of $Ca^{2+}$ and $Mg^{2+}$ and autoclaved to be reused up to a maximum of ten times.

## Cellular apoptosis assay – manual method
After 24 h, media from each well was transferred to microcentrifuge tubes. Cells were detached from the 24-well plate by incubating with 0.25% trypsin-EDTA at 37 °C for 5–6 min. Complete RPMI was added to neutralize the solution and collect the samples from each well to transfer to respective microcentrifuge tubes. Cells were centrifuged at 300 x $g$ for 5 min and washed one time in HBSS with $Ca^{2+}$ and $Mg^{2+}$.

To assess cell apoptosis and necrosis, FITC-conjugated Annexin-V (AV) (BD Pharmingen, San Diego, CA, USA) and propidium iodide (PI) (BD Pharmingen) were used. The manufacturer's instructions were followed to prepare samples for analysis via flow cytometry. Cells were incubated with AV and PI antibodies at room temperature for 15 min in the absence of light, followed by immediate analysis using a Guava easyCyte 12HT benchtop flow cytometer (MilliporeSigma, Burlington, MA, USA). Flow cytometry plots were analyzed using FlowJo software (FlowJo, Ashland, OR, USA).

Viable cells were identified as being negative for AV and PI, necrotic cells were positive for PI, early apoptotic cells were positive for AV, and late apoptotic cells were positive for AV and PI. Control samples were used with each experiment to calibrate the instrument: cell samples labeled with AV or

PI to define boundaries of the cell populations, and unlabeled samples in HBSS to elucidate the level of autofluorescence. The gating used to analyze the AV/PI flow cytometry plots is shown in Fig. S5a, b.

TRAIL sensitization using the viability values (lower left quadrant as shown in Fig. 4e) from the AV/PI assay was calculated using the following equations:

Static + TRAIL:

$$\text{TRAIL Sensitization} = \frac{(\%\text{Cells}) - (\%\text{Cells, TRAIL})}{\%\text{Cells}} \quad (4)$$

FSS + TRAIL:

$$\text{TRAILSensitization(FSS)} = \frac{(\%\text{Cells, FSS}) - (\%\text{Cells, FSS, TRAIL})}{\%\text{Cells, FSS}} \quad (5)$$

For GsMTx-4 treatment, the following equations were used:
Static + TRAIL:

$$\begin{aligned} &\text{TRAIL Sensitization(GsMTx4)} \\ &= \frac{(\%\text{Cells, GsMTx4}) - (\%\text{Cells, TRAIL, GsMTx4})}{\%\text{Cells, GsMTx4}} \end{aligned} \quad (6)$$

FSS + TRAIL:

$$\begin{aligned} &\text{TRAIL Sensitization(GsMTx4 + FSS)} \\ &= \frac{(\%\text{Cells, FSS, GsMTx4}) - (\%\text{Cells, FSS, TRAIL, GsMTx4})}{\%\text{Cells, FSS, GsMTx4}} \end{aligned} \quad (7)$$

## JC-1 assay – manual method
After 24 h, media from each well was transferred to microcentrifuge tubes. Cells were detached from the 24-well plate by incubating with 0.25% trypsin-EDTA at 37 °C for 5–6 min. Complete media was added to neutralize the solution and collect the samples from each well to transfer to respective microcentrifuge tubes.

The manufacturer's directions were followed for the JC-1 (Abcam, Cambridge, MA, USA) assay. Briefly, the cells were centrifuged at 300 x $g$ for 5 min and washed one time in HBSS with $Ca^{2+}$ and $Mg^{2+}$. Samples were then incubated with 10 μM JC-1 dye for 30 min at 37 °C. Cells were washed with HBSS and JC-1 fluorescence was analyzed via flow cytometry using the red-violet and green-violet lasers. Cells with a higher red fluorescence were considered to have healthy mitochondria, and cells with a lower red fluorescence interpreted as having depolarized mitochondria. The gating used to analyze the JC-1 flow cytometry plots is shown in Fig. S5a, c.

## Cellular apoptosis assay and JC-1 analysis – semi-automated method
To determine cellular viability using the VIAFLO96, the following custom VIALINK programs were followed: "AVPI-1.xml" and "AVPI-2.xml". To measure the mitochondrial depolarization using the VIAFLO96, the following VIALINK custom programs were followed: "JC1-1.xml", "JC1-2.xml" and "JC1-3.xml". These programs required the user to standby for minor actions, such as centrifugation steps. The materials required for these methods are listed in Supplementary Data 1, and the programs are provided as supplemental files. The number of pipette tips and reagent volumes used for these semi-automated programs will vary depending on the number of samples. Additional materials required for these programs are the 12 column polystyrene reservoirs (INTEGRA Biosciences) and 300 μL "wide bore" pipette tips (INTEGRA Biosciences).

## Intracellular flow cytometry preparation
To examine changes in cellular response of the BMDCs exposed to shear stress, the following custom VIALINK programs were followed: "FIXED PREP 1.xml", "FIXED PREP 2.xml", "FIXED PREP 3.xml" and "FIXED

PREP 4.xml" (provided as supplemental files). These programs also require the user to standby for centrifugation steps. The materials required for these methods are listed in Supplementary Data 1. The following pre-conjugated antibodies were used: FITC anti-mouse CD11c (N418) (BioLegend, San Diego, CA, USA), PE anti-mouse I-A/I-E (M5/114.15.2) (MHC II) (BioLegend), PE phospho-NF-kB p65 (Ser529) (NFkBp65S529-H3) (Thermo Scientific, Milton Park, UK), PE mouse anti-Ki67 (B56) (BD) and PE anti-mouse CD80 (16-10A1) (BD). The gating used for analysis of the BMDC flow cytometry experiments is shown in Fig. S6.

## Shear stress calculations

MATLAB was used to estimate the time-averaged wall shear stress that a single cell experiences in the narrowing region of a pipette tip at the average fluid velocity. The local wall shear stress was integrated as a trapezoidal rule quadrature. The MATLAB codes are provided as supplemental files. The graphs in Fig. 1 and Figure S2 were prepared using MATLAB.

## Statistics and reproducibility

Data sets were plotted and analyzed for statistical analysis using Prism 10 software (GraphPad, San Diego, CA, USA). All data are reported as the mean and standard error of the mean unless otherwise noted. Two-tailed unpaired $t$-test was used to compare two groups of data and two-way ANOVA test was used to compare multiple groups, with $p < 0.05$ considered significant. At least three independent replicates were used for each experiment.

## Reporting summary

Further information on research design is available in the Nature Portfolio Reporting Summary linked to this article.

## Data availability

The data that support the findings of this study are available within the article and its supplementary material. The source data behind the graphs in the paper can be found in Supplementary Data 2. The data may be made available upon request and the authors request that these data should not be distributed without attribution to this published article.

## Code availability

The MATLAB (version R2023A) and VIALINK (version 5.6.9) codes used in this study are provided as supplemental files, and the authors request that these programs should not be modified or distributed without attribution to this published article. They have also been deposited at Zenodo.org (https://doi.org/10.5281/zenodo.11019126)[80].

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

## Acknowledgements

This research was funded by the United States National Institute of Health grant number R01CA256054 to M.R.K.; and National Science Foundation Graduate Research Fellowship to A.R.F. grant number 2021314768. The illustrations in Figs. 3, 9, and 10a were created by A.R.F. using Microsoft PowerPoint (Version 2403), and some icons (media trough, hand-held multichannel pipette, and pipetting woman) in Figs. 3 and 9 were used from BioRender.

## Author contributions

A.R.F. conceived the studies, responsible for writing. A.R.F. and S.C.R. completed the formal analysis. A.R.F. and M.R.K. developed conceptualization and methodology. A.R.F., S.C.R., S.V.K. and S.J.R. are responsible for the investigation. Mouse procedures were completed by A.R.F., S.J.R. and J.A.D.; M.R.K. is also responsible for supervision and editing.

## Competing interests

The authors declare no competing interests.

## Ethical approval

Ethics approval for experiments reported in this publication on animal subjects was granted and we have complied with all relevant ethical regulations for animal use. The protocol for all mouse studies was approved by the Vanderbilt IACUC protocol #M1700009-02 and mice were monitored by staff from the Division of Animal Care (DAC) at Vanderbilt University.
