## [Peer Review File · Communications Biology]

Reviewers' comments:

Reviewer #1 (Remarks to the Author):

This study presented a high-throughput and semi-automated system to investigate the response of tumor cells to fluid shear stress mimicking shear flow in blood circulation during tumor metastasis. The proposed platform could automate (semi-) the whole process and provide a novel strategy for the field of mechanobiology to explore tumor/immune cell mechanotransduction. Based on this system, the authors replicated the pro-apoptotic effects of TRAIL in sensitizing prostate cancer cells to fluid shear stress, and reported the pro-differentiation effect of shear stress on BMDCs. Several comments are listed below.

1. In the introduction section, another type of current platforms for shear study is peristaltic pump + microfluidic system (e.g., Regmi et al., Sci. Rep. 7, 39975; Xu et al., J Cell Sci. 2022 May 15;135(10):jcs259586; Fan et al., Sci. Rep. 6, 27073;), which should be included for more comprehensive comparison.
2. Several statements should be presented more appropriately. For example, Line 60-61, "This understanding is essential, given that only ~0.01% of CTCs survive in the circulatory system." This might not be true. According to the literature, less than 0.01% of CTCs may eventually grow into macroscopic metastases.
3. In the current system, the shear force experienced by CTCs depends on the radial position within the tubing. In other words, the actual shear stress is still unknown, which warrants further clarification and discussion.

Reviewer #2 (Remarks to the Author):

The manuscript titled 'Multiplex, high-throughput method by study cancer and immune cell mechanotransduction' appears to be a protocol paper suggesting a method for simulating blood flow using a specific pipette tip to provide a physical environment exposing circulating tumor cells or immune cells. The authors state that their developed method is suitable for applying shear stress over an extended period at a rapid pace, addressing limitations in existing shear stress application methods. However, it is noted that for high shear stress, methods involving a peristaltic pump to control speed and time are known, and for low shear stress, methods involving a syringe pump to apply stress over a longer duration have been reported (Li N., Diaz, MF, Wenzel PL, Application of Fluid Mechanical Force to Embryonic Sources of Hemogenic Endothelium and hematopoietic stem cells, Methods Mol Biol, 2015, 183-193).

Furthermore, the paper reports that shear stress, as confirmed in this study, sensitizes cancer cells to TRAIL (tumor necrosis factor-related apoptosis-inducing ligand), a phenomenon previously documented. The paper suggests that it can apply shear stress research by creatively modifying existing methods, indicating that the novelty of the research content itself seems limited.

Reviewer #3 (Remarks to the Author):

In this manuscript by Fabiano et al. titled "Multiplex, high-throughput method to study cancer and immune cell mechanotransduction", the authors describe the modification of a pipetting platform to expose cells to a range of shear forces in a semi-automated and high-throughput fashion. This method is tested in two model systems, the first combining shear forces with TRAIL treatment of prostate cancer cell line showing higher induction of apoptosis, the second showing higher activation marker expression of dendritic cells under fluid shear stress exposure.

The described method is a valuable addition to screen the effect of shear forces on circulating cells, especially because of the possibility for multiplexing. The multiplexing capability is not yet fully implemented yet in this manuscript, but the proof of principle for the approach is established with reference cases based on the groups earlier work. It would be interesting to see cellular responses induced by this Viaflo shear system compared to more traditional approaches, for example by characterizing responses of known mechanotransduction pathways or mRNA profiling. Nonetheless, the method offers exciting opportunities for further exploring interactions between pharmacological treatments and mechanical actuation, and studying the role of shear stress on the immune system.

That said, there are a few typos in the text and I have some concerns with some analyses. I hope the authors can correct or clarify these following points:

0) For a system where mechanotransduction is involved, this approach fills a technology gap, especially in throughput. The measure of throughput of the system can be argued to be less than high. Could the system be scaled to increase throughput?

1) line 119: The Vialink program has been developed "to intracellularly permeabilize". It is unclear to me what distinguishes intracellular permeabilization from regular permeabilization.

2) line 140: typo "mechantransduction"

3) What is the effective volume exposed to the shear per pipetting cycle, is there a dead volume on the inside/outside of the tip that does not cross the narrow region of max FSS?

4) line 196-198. The importance of this statement is not immediately obvious to me.

5) line 214: a  as.

6) Line 223: considering "due to their lymph node origin", I am not sure I follow the logic here. If the LNCaP cells are of lymph node origin, wouldn't they be more accustomed to shear stresses, vs the PC3 cells? I am interpreting this as that the LNCaP cells die sooner (are they more sensitive or less resistant?), and your referenced earlier work seems to confirm that (Hope et al 2021). Perhaps it is clearer here to not couple this to lymph node origin per se, but simply to your earlier findings. If you have more examples of lymph node derived lines that are less resistant to shear, this case could be stronger, but I don't think it's a main point of the manuscript.

7) line 234: "Thought" seems the wrong word here

8) Fig. 4: in the static case, what is on the x-axis, since there's no shear cycles.

9) line 270: In my view these results don't really encompass how TRAIL therapeutic effects are enhanced, but describe more that they are enhanced. A molecular characterization of the signaling involved would describe how that is happening.

10) I would also be interested in the total cell numbers that survive the procedure, not just the fraction of viable/apoptotic/necrotic cells after 24 hours post shear, to get an idea of the harshness of the shear exposure. Is it possible to report the numbers of cells that enter the experiment vs the number of cells that remain at the end?

11) line 279: Depolarization was most significant compared to what? I think what's more interesting is that %depolarization is highest in these cases.

12) Line 274 and 295: the use of "intrinsically" is confusing to me. TRAIL signaling also induces apoptosis without shear, how can piezo1 then be an intrinsic part of the pathway. It does appear synergistic, but not intrinsic.

13) figure 5: It is not obvious which contrast the different colors of significance stars correspond to. Perhaps a table with contrasts for the statistical analyses would be helpful.

14) Figure 5: the figure legend states 'mitochondrial dysfunction', however it can be argued the mitochondrion is doing what it should do, responding to signaling cues and inducing apoptosis. Perhaps 'mitochondrial depolarization' is a better descriptor of the figure.

15) Figure 4a-d: these were analyzed with unpaired t-tests, but in this case with multiple variables and a possible interaction I would think a 2-way ANOVA is the correct analysis.

16: Fig 6: here a 2-way ANOVA was performed, did that reveal an interaction term between FSS and TRAIL?

17: line 330: Reference to fig 6d should be to 6c.

18: line 342: The term MSC and Piezo1 are used quite interchangeably, where it seems the authors prefer to attribute the findings to Piezo1. However GSMTx-4, to my knowledge, is not that specific, and for this specific manuscript it doesn't seem to be so important which MSC it is.

19: Fig 7: Despite the fold change, the conditions FSS + TRAIL vs FSS + TRAIL + GSMTX-4 are not significantly different. Perhaps there are other MSC pathways involved that are not inhibited by GSMTX4.

20: Fig 8: The figure doesn't show greater degree of mechanoactivation, but greater degree of TRAIL sensitization/effectiveness with 22G tips. For mechanoactivation specific mechanotransductory pathways would need to be analyzed.

21: Fig 9a caption: is this needle still a 22G needle?

22: line 481: for 'invariability' consider 'consistency'

23: Equation 3: the equation is not intuitive to me, which quadrants of the FACS plots are subtracted here? And I would expect with the 'sensitization' term for there to be a comparison between static and shear in the formula. This seems more like a formula for effectiveness. This is also reflected in e.g. fig. 4h, where the y axis is the TRAIL effectiveness, and the slope of the curve could be described as the 'sensitization'.

24: The title states the multiplex aspect of the system, however from the text there is little content on what is exactly meant by this multiplex aspect. Perhaps it would add strength to the manuscript to make this aspect more concrete.

Reviewer #1 (Remarks to the Author):

This study presented a high-throughput and semi-automated system to investigate the response of tumor cells to fluid shear stress mimicking shear flow in blood circulation during tumor metastasis. The proposed platform could automate (semi-) the whole process and provide a novel strategy for the field of mechanobiology to explore tumor/immune cell mechanotransduction. Based on this system, the authors replicated the pro-apoptotic effects of TRAIL in sensitizing prostate cancer cells to fluid shear stress, and reported the pro-differentiation effect of shear stress on BMDCs.

Several comments are listed below.

1. In the introduction section, another type of current platforms for shear study is peristaltic pump + microfluidic system (e.g., Regmi et al., Sci. Rep. 7, 39975; Xu et al., J Cell Sci. 2022 May 15;135(10):jcs259586; Fan et al., Sci. Rep. 6, 27073;), which should be included for more comprehensive comparison.

Thank you for this comment, we have addressed these methods as well and a sentence about the drawbacks associated with them along with these relevant references. Lines 58 – 60 and 80 – 85.

2. Several statements should be presented more appropriately. For example, Line 60-61, "This understanding is essential, given that only ~0.01% of CTCs survive in the circulatory system." This might not be true. According to the literature, less than 0.01% of CTCs may eventually grow into macroscopic metastases.

You are correct. We have changed this wording to be more precise, by stating “less than 0.01% of CTCs survive in the circulatory system” for clarification in the Introduction and in the Abstract.

3. In the current system, the shear force experienced by CTCs depends on the radial position within the tubing. In other words, the actual shear stress is still unknown, which warrants further clarification and discussion.

Our estimate is based on modeling CTCs through the circulation as experiencing laminar, Hagen-Poiseuille flow through the circulatory system. We have added a clarification on this point as suggested (lines 182 – 187). We have also added a calculation to the Methods section to model the minimum FSS that a cell would experience in the 22 G needle region, based on the radius of the PCa cells (lines 594 – 595).

Reviewer #2 (Remarks to the Author):

The manuscript titled 'Multiplex, high-throughput method by study cancer and immune cell mechanotransduction' appears to be a protocol paper suggesting a method for simulating blood flow using a specific pipette tip to provide a physical environment exposing circulating tumor cells or immune cells. The authors state that their developed

method is suitable for applying shear stress over an extended period at a rapid pace, addressing limitations in existing shear stress application methods.

However, it is noted that for high shear stress, methods involving a peristaltic pump to control speed and time are known, and for low shear stress, methods involving a syringe pump to apply stress over a longer duration have been reported (Li N., Diaz, MF, Wenzel PL, Application of Fluid Mechanical Force to Embryonic Sources of Hemogenic Endothelium and hematopoietic stem cells, Methods Mol Biol, 2015, 183-193).

Thank you for this comment. We have added additional references as suggested in the Introduction that use microfluidic chips coupled with syringe pumps to provide laminar flow through the system (lines 80 – 85). These methodologies differ from those that we present in our manuscript. Although consisting of a syringe pump, previous studies model low-intensity FSS (~5 dyn/cm²) and low flow rates, significantly lower than those presented in our modified system. Additionally, the VIAFLO96 methods presented in our manuscript use a commercially available device that is not difficult to set-up (whether the methods are employed with or without the modified 22 G needle-tips). However, microfluidic chips, whether used with a peristaltic or syringe pump, are more complex to develop, requiring time and labor from researchers and these systems are not as high-throughput (meaning, few conditions can be tested at once) relative to our system.

Furthermore, the paper reports that shear stress, as confirmed in this study, sensitizes cancer cells to TRAIL (tumor necrosis factor-related apoptosis-inducing ligand), a phenomenon previously documented. The paper suggests that it can apply shear stress research by creatively modifying existing methods, indicating that the novelty of the research content itself seems limited.

We do not propose to necessarily modify existing methods for its own sake, but provide a new method using a commercially available device to recreate physiological FSS in vitro, while overcoming limitations associated with other developed methods (viscometers, syringe pumps, etc.) as described in the Introduction and Discussion regarding the relatively low number of conditions those single-channel systems can assay at one time. We also provide examples in the Discussion regarding ways in which the methodologies developed in this manuscript could be further tailored and used in combination with additional downstream assays, such as Seahorse, ELISAS, etc., which were beyond the scope of these initial studies to test the efficacy of the system in development. Additionally, we demonstrate the versatility and effectiveness of these methods in Figure 9 & 10, by using the modified system to study the effects of physiological FSS to activate primary, murine immune cells that were subsequently processed in a multiplex manner to be fixed, permeabilized and stained for flow cytometry (targeting multiple activation & co-stimulatory molecules at once). We clarified the method proposed here to administer FSS is a new/ alternate method in the Introduction (line 121) and Discussion (lines 469 – 471, 547 – 553) in ways that overcome limitations with previously established laboratory techniques.

Reviewer #3 (Remarks to the Author):

In this manuscript by Fabiano et al. titled "Multiplex, high-throughput method to study cancer and immune cell mechanotransduction", the authors describe the modification of a pipetting platform to expose cells to a range of shear forces in a semi-automated and high-throughput fashion. This method is tested in two model systems, the first combining shear forces with TRAIL treatment of prostate cancer cell line showing higher induction of apoptosis, the second showing higher activation marker expression of dendritic cells under fluid shear stress exposure. The described method is a valuable addition to screen the effect of shear forces on circulating cells, especially because of the possibility for multiplexing. The multiplexing capability is not yet fully implemented yet in this manuscript, but the proof of principle for the approach is established with reference cases based on the groups earlier work. It would be interesting to see cellular responses induced by this Viaflo shear system compared to more traditional approaches, for example by characterizing responses of known mechanotransduction pathways or mRNA profiling. Nonetheless, the method offers exciting opportunities for further exploring interactions between pharmacological treatments and mechanical actuation, and studying the role of shear stress on the immune system.

That said, there are a few typos in the text and I have some concerns with some analyses. I hope the authors can correct or clarify these following points:

0) For a system where mechanotransduction is involved, this approach fills a technology gap, especially in throughput. The measure of throughput of the system can be argued to be less than high. Could the system be scaled to increase throughput?

Thank you for these comments. The multiplex, high-throughput capabilities of this technology could indeed be increased beyond what is presented here, and the versatility and scalability associated with using a semi-automated 96-channel pipetting device is stated in a couple places in the text. For instance, in the discussion section we point out that "these protocols can be adapted to other downstream quantitative assays including ELISAs, Seahorse, luciferase reporter assays..." in the Discussion. Additional models of this device are available, such as the VIAFLO384, in which the device can operate with a 384-well plate to scale up these studies even more – as suggested by the reviewer, we have added a new statement in the Discussion section (lines 547 – 553) to address this point. Additionally, the process overview shown in Fig. 3 is an example of a protocol that can be used with this method shown for simplicity, which can easily be scaled up to contain samples that occupy the entire 96-well plate.

1) line 119: The Vialink program has been developed "to intracellularly permeabilize". It is unclear to me what distinguishes intracellular permeabilization from regular permeabilization.

This a good point and we have removed the word "intracellularly" here (line 114). We were attempting to state that we can intracellularly stain following permeabilization & fixation using this device, but it indeed could be confusing.

2) line 140: typo "mechantransduction"

Thank you, this has been corrected.

3) What is the effective volume exposed to the shear per pipetting cycle, is there a dead volume on the inside/outside of the tip that does not cross the narrow region of max FSS?

The volume programmed to be exposed to the shear per each pipetting cycle is 190 μL to account for minimal volume loss within each well that may arise for the longer duration shear exposures. During a single shearing cycle (i.e. one aspiration + one dispense), 190 μL of cell suspension is aspirated and 190 μL of cell suspension is dispensed, so that the entire volume is exiting the narrow region of the 22 G needle-tips each time. We have added a statement to clarify this point, on lines 226 - 228.

4) line 196-198. The importance of this statement is not immediately obvious to me.

Thank you, we have removed this sentence to reduce confusion since it is not necessary.

5) line 214: a  as.

This has been corrected.

6) Line 223: considering "due to their lymph node origin", I am not sure I follow the logic here. If the LNCaP cells are of lymph node origin, wouldn't they be more accustomed to shear stresses, vs the PC3 cells? I am interpreting this as that the LNCaP cells die sooner (are they more sensitive or less resistant?), and your referenced earlier work seems to confirm that (Hope et al 2021). Perhaps it is clearer here to not couple this to lymph node origin per se, but simply to your earlier findings. If you have more examples of lymph node derived lines that are less resistant to shear, this case could be stronger, but I don't think it's a main point of the manuscript.

We have edited this to reference previous work in our lab that has shown the LNCaP cells to be more sensitive to mechanical stimulus, specifically, high-intensity fluid shear stress (lines 220 – 224). In that prior study there is an apparent correlation between syngeneic cancer cell lines, their relative mechanoresistance, and from what organ they were originally harvested from.

7) line 234: "Thought" seems the wrong word here

This has been corrected.

8) Fig. 4: in the static case, what is on the x-axis, since there's no shear cycles.

These static samples remained on the rotator for the duration of shearing cycles that were tested to serve as control conditions. The x-axis is the same for the static samples. For example, if PCa cells were exposed to 2,000 shearing cycles (78 min), the static samples were

on the rotator for 78 min during the experiment. This applies for Figures 5 – 8, 10 as well. We have added a statement related to this in the Methods section to improve clarity (line 678 – 679).

9) line 270: In my view these results don't really encompass how TRAIL therapeutic effects are enhanced, but describe more that they are enhanced. A molecular characterization of the signaling involved would described how that is happening.

We have changed “how” to “that” to clarify that this present study is reporting the pro-apoptotic effects of TRAIL to be enhanced, without considering any mechanistic signaling involved here (lines 276 – 277). The molecular mechanism of this enhancement was more fully explored in our prior work (e.g., JM Hope et al., Cell Death & Disease, 2021).

10) I would also be interested in the total cell numbers that survive the procedure, not just the fraction of viable/apoptotic/necrotic cells after 24 hours post shear, to get an idea of the harshness of the shear exposure. Is it possible to report the numbers of cells that enter the experiment vs the number of cells that remain at the end?

It is possible to report the number of shear cells that enter the experiments vs those that remain at the end of the experiment by cell counting assays. However, we did not record these values for the studies completed here. We have added a supplemental figure (Figure S3) to the Supplemental Information doc to show representative brightfield images of the LNCaP and PC3 cells after shear exposure (5,000 shearing cycles) with or without TRAIL as a visual demonstration to illustrate how the cell count changes and also to depict cell clustering that is observed for the LNCaP cells post-shear. We have added a statement in lines 265 – 266 to mention this, as well as a statement in the Methods section (lines 690 – 692).

Fig S3 shown below:

11) line 279: Depolarization was most significant compared to what? I think what's more interesting is that %depolarization is highest in these cases.

This is correct, we have changed “most significant” to “highest.” (line 286)

12) Line 274 and 295: the use of "intrinsically" is confusing to me. TRAIL signaling also

induces apoptosis without shear, how can piezo1 then be an intrinsic part of the pathway. It does appear synergistic, but not intrinsic.

We have rephrased this section to improve clarity (lines 280 – 283 and lines 300 – 302). DR4/5 recognizing TRAIL represents the extrinsic apoptotic pathway, whereas Piezo1 activation leading to mitochondrial depolarization represents the intrinsic apoptotic pathway before these pathways lead to the same execution of late-stage apoptosis. Piezo1 activation indicates activation of the intrinsic pathway due to the observed mitochondrial depolarization.

13) figure 5: It is not obvious which contrast the different colors of significance stars correspond to. Perhaps a table with contrasts for the statistical analyses would be helpful.

We have clarified the legend in Fig. 4 & 5. We have also added tables to the supplemental materials to specify that all significance comparisons correspond to the following since we cannot show all the comparisons on the xy-plots without affecting their readability:

Figure 4 – Tables S6 & S7 (AV/PI)

Figure 5 – Tables S8 & S9 (JC-1)

These tables are in the excel sheet “Supplementary Data (Tables S6-S12).”

14) Figure 5: the figure legend states 'mitochondrial dysfunction', however it can be argued the mitochondrion is doing what it should do, responding to signaling cues and inducing apoptosis. Perhaps 'mitochondrial depolarization' is a better descriptor of the figure.

This is a fair point, and we have edited this section to state “depolarization” instead of “dysfunction”.

15) Figure 4a-d: these were analyzed with unpaired t-tests, but in this case with multiple variables and a possible interaction I would think a 2-way ANOVA is the correct analysis.

As suggested by the reviewer, we have re-analyzed this comparison using two-way ANOVA, and adjusted the significance on the plots (a-d) as appropriate. We have updated the figure legend to clarify precisely what the colored stars refer to. We have also added supplemental tables (as mentioned in comment 14) showing two-way ANOVA significance assessment from GraphPad prism since xy-graphs do not allow for all of these comparisons to be clearly indicated. While updating these graphs, we also updated some of the data, evident by minor shifts in some of the xy-lines and added tick marks to all of the representative flow cytometry plots.

We have also reanalyzed the plots in Figure 5a,b using two-way ANOVA for consistency.

Updated Figure 4:

Updated graphs in Figure 5:

16: Fig 6: here a 2-way ANOVA was performed, did that reveal an interaction term between FSS and TRAIL?

The two-way ANOVA did reveal that differences were observed between the FSS-only and TRAIL-only (static) conditions. To convey this, we have added table S11 to show in detail the interactions between the static and shear groups (a-c) to avoid showing too many interactions on the graphs themselves. Table S11 to the "Supplemental Data (Tables S6-S12)" with a complete analysis as now stated in the figure caption.

We also updated these bar graphs (a-c, e) to show all the individual data points, and updated some of the significance bars shown while doing so to further demonstrate the relationship between the static and shear samples and show consistency with the previous results in Figure 4 & 5.

17: line 330: Reference to fig 6d should be to 6c.

This has been corrected, thank you.

18: line 342: The term MSC and Piezo1 are used quite interchangeably, where it seems the authors prefer to attribute the findings to Piezo1. However GSMTx-4, to my knowledge, is not that specific, and for this specific manuscript it doesn't seem to be so important which MSC it is.

We have rewritten this text to state: "To confirm MSC activation by FSS, such as Piezo1" (line 353 – 355). GsMTx-4 does inhibit Piezo1 and TRPV channels, however there is no other known specific inhibitor of Piezo1 (or cationic MSCs).

19: Fig 7: Despite the fold change, the conditions FSS + TRAIL vs FSS + TRAIL + GSMTX-4 are not significantly different. Perhaps there are other MSC pathways involved that are not inhibited by GSMTX4.

We agree, and have added a sentence to convey this idea (lines 371 – 373) with references.

In Figure 7, we reformatted the bar plots to show individual data points (c -f) and updated some of the significance bars that are shown. Formatted all flow cytometry plots to show tick marks.

20: Fig 8: The figure doesn't show greater degree of mechanoactivation, but greater degree of TRAIL sensitization/effectiveness with 22G tips. For mechanoactivation specific mechanotransductory pathways would need to be analyzed.

We have rephrased this title to state: "22 G needles induce higher pro-apoptotic effects of TRAIL compared to unmodified INTEGRA pipette tips" in the text accompanying Figure 8.

Updated the bar graphs in Figure 8 (a,b,d,e) to show individual data points and updated some of the significance markers shown to expand on the statistical relationship between the static and shear samples. Formatted all flow cytometry plots to show tick marks.

21: Fig 9a caption: is this needle still a 22G needle?

Yes it is, and we have attempted to clarify this detail in the text.

22: line 481: for 'invariability' consider 'consistency'

This has been changed to “consistency”.

23: Equation 3: the equation is not intuitive to me, which quadrants of the FACS plots are subtracted here? And I would expect with the 'sensitization' term for there to be a comparison between static and shear in the formula. This seems more like a formula for effectiveness. This is also reflected in e.g. fig. 4h, where the y axis is the TRAIL effectiveness, and the slope of the curve could be described as the 'sensitization'.

The viability plots from the AV/PI assay are those from the lower left quadrant of the flow plots, once the data are analyzed as shown in Fig. 4e (line 723). The slope of the curve in Fig. 4h does confirm our observation that increasing the shearing exposure of the PCa cells leads to an increase in the TRAIL sensitization, as mentioned on line 261, however we don't necessarily use that slope as our TRAIL sensitization value.

TRAIL sensitization equations have been developed and published previously in our lab's publications. We have retained the previously used FSS+TRAIL sensitization equation to consist of only viability values from the FSS condition for consistency, so that the effects of solely the FSS can be assessed and compared with prior results. If we compared the viability of the FSS+TRAIL condition to the static-only condition, then we cannot completely rule out the effects that FSS has on enhancing the apoptotic effects of TRAIL.

We have added equations to clarify the common usage of the TRAIL sensitization equations for given conditions (equations 4 – 7) in the Methods section (lines 724 – 734).

In Figure 4h, we do not directly compare the static and shear samples since we are primarily interested in confirming that the TRAIL sensitization is dependent on the shear exposure at a constant value of TRAIL concentration.

Here are some of our lab's prior work where we employ a similar quantification of TRAIL sensitization:

Knoblauch et al., *ACS Omega* 2023, 8,16975–16986.

Hope et al., *Cell Death and Disease* (2019)10:837.

Michael J Mitchell and Michael R King, 2013 *NewJ.Phys.* 15 015008.

24: The title states the multiplex aspect of the system, however from the text there is little content on what is exactly meant by this multiplex aspect. Perhaps it would add strength to the manuscript to make this aspect more concrete.

As suggested, we have elaborated on the multiplex capabilities more in the Introduction and Discussion. The multiplex capabilities also refer to the overall multiwell format that this device and these protocols provide for studies of shear stress mechanotransduction. Since this is the first report that uses our modified VIAFLO96 system, the multiplex capabilities also correspond to future applications that the device may be integrated with.

Other edits by the authors:

- The authors completed the Nature Review checklist.
- We shortened the abstract to meet the 150 word limit.
- We moved the supplemental figures and small tables to the document titled “Supplementary Information” and the previous Tables S6 & S7 to the “Supplementary Data (Tables S6-S12)” excel file.
- The new tables that were added to “Supplementary Data (Tables S6-S12)” are highlighted in the excel sheet.
- Created a new supplementary excel sheet → “Supplementary Data (Source Data)” to include all of the raw data shown in the graphs in the manuscript, and updated some data/graphs while consolidating all of the data.
- Edited all bar plots to show all points, and make some minor modifications to update some graphs and data while doing so.
- We have added the numbers (scales) to the axis of the lower left representative flow cytometry plots in all the figures, and have added ticks on the x- and y-axis of all flow plots shown.
- Figure 10:
 - Fig. 10b → changed data to box and whisker plots, specified in lines 434 – 435, with the median shown as (+).
 - Fig. 10c → all flow cytometry plots have ticks shown on axis now.

Edits to supplemental figures (“Supplementary Information”):

- Figure S3 was added (in response to reviewer comment 10).
- Figure S4 → updated, flow plots show tick marks.

- Figure S5 → added to show flow cytometry gating for cancer cell experiments.

- Figure S6 → added to show flow cytometry gating for BMDc experiments.

REVIEWERS' COMMENTS:

Reviewer #1 (Remarks to the Author):

Thank the authors for the efforts in addressing most of the comments. A minor point: According to the literature, less than 0.01% of CTCs may eventually grow into macroscopic metastases. However, the authors stated that "less than 0.01% of CTCs survive in the circulatory system", which is again incorrect. Since there are various rate-limiting factors in every step of metastasis, including circulation, less than 0.01% of CTCs that survive in the circulatory system should lead to much lower rate of metastatic colonization. The authors should correct this before it can be published.

Reviewer #2 (Remarks to the Author):

I have thoroughly reviewed the rebuttal letter and the revised manuscript. All comments or concerns have been adequately addressed. I am pleased with the revisions made to the manuscript.

Reviewer #3 (Remarks to the Author):

I thank the authors for their clarifications and corrections. All my concerns have been addressed.

Reviewer #1 (Remarks to the Author):

Thank the authors for the efforts in addressing most of the comments. A minor point: According to the literature, less than 0.01% of CTCs may eventually grow into macroscopic metastases. However, the authors stated that “less than 0.01% of CTCs survive in the circulatory system”, which is again incorrect. Since there are various rate-limiting factors in every step of metastasis, including circulation, less than 0.01% of CTCs that survive in the circulatory system should lead to much lower rate of metastatic colonization. The authors should correct this before it can be published.

Thank you for the correction. The authors have rephrased to clarify that “less than 0.01% of CTCs eventually may grow into macroscopic metastases.”

Reviewer #2 (Remarks to the Author):

I have thoroughly reviewed the rebuttal letter and the revised manuscript. All comments or concerns have been adequately addressed. I am pleased with the revisions made to the manuscript.

Thank you for your feedback and time reviewing this manuscript!

Reviewer #3 (Remarks to the Author):

I thank the authors for their clarifications and corrections. All my concerns have been addressed.

Thank you for your feedback and time reviewing this manuscript!